# HKAN: Hierarchical Kolmogorov-Arnold Networks for Efficient and Interpretable Feature Interaction Modeling

## Abstract

Learning complex feature interactions is central to modern machine learning, driving breakthrough performance across domains from structured data analytics to predictive modeling in recommender systems and beyond. However, despite notable progress, this field still faces three substantial challenges: i) lack of adaptive topology discovery — models cannot automatically learn which features should interact and at what order; ii) the 'black-box' nature of deep neural networks with poor explainability of the learned interaction patterns; iii) computational inefficiency due to parameter-heavy architectures with limited scalability. To address these challenges, we propose a hierarchical sparse framework, namely Hierarchical Kolmogorov-Arnold Network (HKAN), for efficient and interpretable feature interaction modeling with three key aspects: i) factor-quality-guided evolutionary architecture search (FG-EAS) to automatically discover data-centric optimal feature grouping strategies; ii) hierarchical sparse structure with superior parameter efficiency iii) B-spline-based univariate function visualization and hierarchical factor structures with end-to-end interpretability from local to global levels. To test the predictive and symbolic regression ability of HKAN, we conduct experiments across 10 tabular learning and 2 function fitting tasks. HKAN achieves state-of-the-art (SOTA) or highly competitive performance on the vast majority of datasets while utilizing significantly fewer parameters. Notably, on three of these datasets, it reaches state-of-the-art performance with less than 10% of the parameters used by the baseline models. Moreover, HKAN can serve as a knowledge discovery tool with excellent explainability (e.g., explicit formulas of data patterns) compared to other black-box baselines, which represents a significant step toward building more trustworthy and accountable AI systems.

## 1 Introduction

Learning complex feature interactions is a core capability of modern deep learning (Goodfellow et al., 2016), driving breakthrough performance across domains from structured data analytics (Guo et al., 2017) to predictive modeling (Covington et al., 2016). While this capability has enabled remarkable advances, current approaches face three fundamental challenges that limit their practical deployment. First, they lack **adaptive topology discovery** — models like xDeepFM (Lian et al., 2018) enforce rigid architectural constraints (e.g., FM is strictly limited to 2nd-order interactions, xDeepFM's interaction order is fixed by predefined depth), while attention-based models like FT-Transformer (Gorishniy et al., 2021) assume dense all-to-all connectivity, failing to explicitly isolate sparse interaction subsets. These rigid assumptions prevent automatic adaptation to dataset-specific interaction patterns. Second, the **black-box nature** of deep models precludes understanding of learned interactions (Ribeiro et al., 2016), which is unacceptable in high-stakes domains where interpretability is crucial (Bodria et al., 2023). Third, **computational inefficiency** plagues existing methods — FT-Transformer requires 70K+ parameters even for small datasets, while TabNet exceeds 450K — leading to overfitting risks and limiting deployment in resource-constrained environments (Grinsztajn et al., 2022).

Inspired by recent advances in Kolmogorov-Arnold Networks (KANs) (Liu et al., 2024), we propose the Hierarchical Kolmogorov-Arnold Network (HKAN), an evolutionary search-driven framework

designed to resolve the fundamental three-way challenge of achieving simultaneous **automated topology discovery**, **intrinsic interpretability**, and **parameter efficiency**. Its core innovation is a sparse, two-level hierarchical architecture that decomposes complex global interactions: multiple lightweight KANs first process semantically related feature subsets into interpretable 'factors', which are then modeled by a global KAN, ensuring both parameter efficiency and a transparent, multi-level interpretive path. Critically, this architecture is not manually designed but discovered automatically by our novel factor-quality-guided evolutionary architecture Search (FG-EAS), a method that moves beyond traditional performance-only optimization by co-optimizing for both predictive accuracy and the explicit quality of the learned representation.

Our comprehensive evaluation across diverse tabular benchmarks demonstrates HKAN's remarkable capabilities in achieving the elusive combination of automated topology discovery, intrinsic interpretability, and parameter efficiency. On UCI Heart Disease (Asuncion et al., 2007), HKAN achieves superior performance with only 1.7K parameters—outperforming FT-Transformer (70K parameters) while using 97% fewer parameters. This superior parameter efficiency (90-99% reduction compared to existing deep learning methods) is consistent across all benchmarks while maintaining or exceeding baseline performance. Beyond predictive tasks, HKAN excels as a knowledge discovery tool: on function fitting tasks, it accurately identifies true feature dependencies and provides transparent symbolic expressions that remain hidden to black-box models, establishing its value for understanding complex data patterns.

## 2 RELATED WORK

**Feature Interaction Modeling.** The evolution of feature interaction modeling reflects a progression from simple pairwise to complex high-order interactions. Early work with Factorization Machines (Rendle, 2010) demonstrated that even second-order interactions could significantly improve performance, inspiring a series of extensions. To capture higher-order patterns, researchers developed two parallel paths: explicit interaction modeling through cross-networks (Wang et al., 2017) and compressed interactions (Lian et al., 2018), and implicit modeling through deep neural networks (Guo et al., 2017). The latest generation combines both strategies—AutoInt (Song et al., 2019) uses multi-head self-attention to automatically detect relevant interactions, while models like xDeepFM (Lian et al., 2018) jointly train explicit and implicit components. However, all these approaches share a fundamental limitation: they cannot automatically discover which specific feature subsets should interact. FM is restricted to pairwise interactions, xDeepFM's interaction order is fixed by predefined network depth, and attention-based models assume dense all-to-all connectivity, failing to identify sparse, semantically meaningful feature groupings. Recent sparse high-order methods like BFIS (**?**) and iRF (**?**) attempt to address this, but face critical constraints: BFIS requires manually presetting maximum interaction order and suffers exponential complexity ($\mathcal{O}(|\mathcal{F}|^H K)$), while iRF detects feature subsets via tree-based methods but produces non-smooth step functions unsuitable for precise mathematical modeling.

**Tabular Data as a Testbed.** The tabular domain has become the primary testbed for feature interaction methods due to its unique characteristics: heterogeneous features, irregular patterns, and clear interpretability requirements (Shwartz-Ziv & Armon, 2022). This has led to specialized architectures—TabNet (Arik & Pfister, 2021) adds sequential decision-making, SAINT (Somepalli et al., 2021) incorporates inter-sample attention, and FT-Transformer (Gorishniy et al., 2021) treats features as tokens. Recent work like TabKANet (Gao et al., 2024) explores KAN for tabular tasks but maintains dense connectivity. While our experiments follow this tradition of tabular evaluation for rigorous comparison, HKAN's core innovation—automated hierarchical decomposition—extends beyond tabular data to any domain where feature interactions matter.

**Kolmogorov-Arnold Networks (KANs) and Extensions.** KANs (Liu et al., 2024) replace MLPs' linear weights with learnable B-spline activation functions, enabling direct visualization and symbolic extraction of learned patterns. This interpretability advantage has inspired extensions across domains: TKAN (Genet & Inzirillo, 2025) for time series, KAN-Transformer (Xingyi Yang, 2025) for sequence modeling, GKAN (Kiamari et al., 2024) for graph neural networks, and explorations in computer vision (Mohan et al., 2024). However, these variants inherit KAN's quadratic parameter scaling, limiting their practical applicability to high-dimensional problems.

**Interpretable Models and Post-hoc Methods.** The pursuit of interpretability has led to two distinct paradigms. Intrinsic interpretable models like NODE-GAM (Chang et al., 2022) and EBM (Nori et al., 2019) build transparency into their architecture—NODE-GAM uses oblivious decision trees to construct generalized additive models with shape plots for visual inspection. While effective for understanding feature effects, these tree-based approaches produce non-smooth step functions that cannot be easily converted to explicit mathematical formulas. In contrast, post-hoc methods like Zhang et al. (Zhang et al., 2022) analyze trained black-box models to detect interactions after training. These approaches require first training a dense model, then performing secondary analysis to identify interaction patterns, which neither reduces the computational cost of the original model nor provides symbolic mathematical expressions. HKAN bridges these paradigms by combining intrinsic interpretability through B-spline-based architecture with the capability to extract explicit symbolic formulas, while simultaneously discovering sparse interaction topology through evolutionary search.

## 3 METHODOLOGY

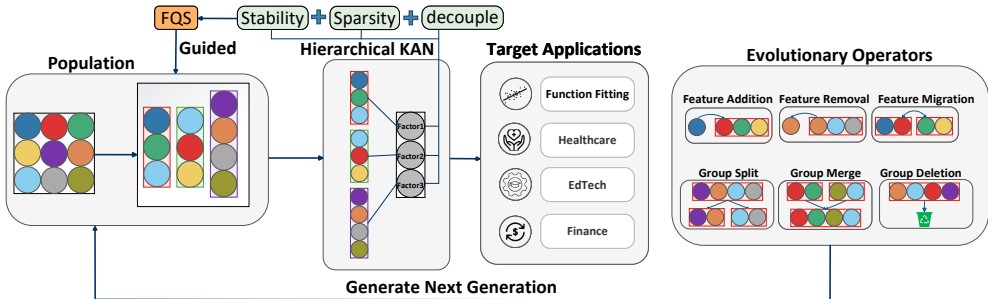

Figure 1: Overview of HKAN architecture. The framework combines evolutionary feature grouping (first panel) with hierarchical KAN processing (second panel) to enable Versatile Applications (third panel). The rightmost component shows the six mutation operators used in evolutionary search. FQS guides the evolutionary process to discover optimal feature groupings that produce high-quality semantic factors.

### 3.1 PROBLEM FORMULATION AND HKAN OVERVIEW

We formalize this challenge in two complementary settings that showcase HKAN's versatility. For **tabular data prediction**, we learn $f : \mathbb{R}^n \to \mathbb{R}^m$ from dataset $\mathcal{D} = \{(\mathbf{x}_i, y_i)\}_{i=1}^N$ to minimize prediction error while maintaining interpretability. For **function fitting**, we seek to recover both the predictive mapping and the explicit symbolic form $\hat{f}(x) \approx f^*(x)$ from observations, enabling scientific insight into underlying relationships. Both settings share a common challenge: discovering how features naturally group and interact, which traditional approaches address through manual architecture design (Wang et al., 2017; Lian et al., 2018).

HKAN introduces an integrated architecture consisting of three tightly coupled components that work synergistically to address this challenge. First, the **hierarchical sparse architecture** transforms dense KAN into an efficient sparse structure with overlapping feature groups, dramatically reducing parameter complexity while preserving expressiveness. Second, **dual-layer regularization** coordinates constraints on both B-splines and semantic factors to ensure learned representations are both accurate and interpretable. Third, **factor-quality-guided evolution** automatically discovers optimal feature groupings through evolutionary search guided by explicit quality metrics measuring independence, sparsity, and stability. These components form an end-to-end system where architecture discovery, model training, and interpretability constraints are jointly optimized.

As illustrated in Figure 1, these components operate in an integrated pipeline. The evolutionary search (first panel) explores different feature grouping strategies, evaluating each through the Factor Quality Score that measures independence, sparsity, and stability. Selected architectures instantiate

hierarchical KANs (second panel) where mini-KANs process overlapping feature groups to extract semantic factors, which are then integrated by a global KAN. During training, dual-layer regularization ensures that both the learned B-spline functions and extracted factors maintain interpretability. This end-to-end optimization produces models that achieve state-of-the-art performance while providing transparent insights into feature relationships.

We now detail how these three components—hierarchical architecture, dual-layer regularization, and evolutionary search—work together to achieve efficient and interpretable feature interaction modeling.

## 3.2 HIERARCHICAL SPARSE ARCHITECTURE WITH OVERLAPPING GROUPS

The original KAN's dense connectivity leads to quadratic parameter scaling that severely limits its applicability to high-dimensional problems. Moreover, treating all features uniformly ignores their natural semantic groupings and multifaceted roles in different contexts. We address both challenges through a hierarchical sparse design with overlapping feature groups.

The original Kolmogorov-Arnold Network (Liu et al., 2024) is grounded in the Kolmogorov-Arnold representation theorem (Kolmogorov, 1957), which posits that any continuous function $f$ for $n \geq 2$ can be decomposed into a dense representation:

$$f(x_1, x_2, \ldots, x_n) = \sum_{q=0}^{2n} \Phi_q \left( \sum_{p=1}^{n} \psi_{p,q}(x_p) \right) \tag{1}$$

While theoretically powerful, this dense connectivity hinders practical application. To address this, HKAN proposes a structured sparse alternative, reframing the function as a factorized decomposition over semantically grouped features:

$$f(x_1, \ldots, x_n) = \sum_{k=1}^{K} \Phi_k \left( \sum_{i \in G_k} \psi_{i,k}(x_i) \right) \tag{2}$$

where $G_k \subset \{1, 2, \ldots, n\}$ are feature groups and $K \ll 2n + 1$. This decomposition naturally leads to our overlapping group structure, formalized through a binary assignment matrix $\mathbf{M} \in \{0, 1\}^{n \times K}$ where $M_{ik} = 1$ indicates feature $i$ belongs to group $k$.

A key innovation is permitting overlaps between groups ($G_i \cap G_j \neq \emptyset$), allowing features to participate in multiple semantic contexts. This design captures the reality that features often play multifaceted roles—for instance, in medical diagnosis, age may indicate both risk factors and recovery potential. Formally, we define $\text{KAN}_k : \mathbb{R}^{|G_k|} \to \mathbb{R}$ as the sub-network parameterized by learnable B-spline functions on edges defined by group topology $G_k$, and $\text{Factor}_k \in \mathbb{R}^{N \times 1}$ as the latent semantic representation output by the $k$-th group for all $N$ samples in the batch. Unlike traditional disjoint partitioning (Song et al., 2019), each mini-KAN processes its assigned features to produce interpretable factors:

$$\text{Factor}_k = \text{KAN}_k(\{x_i : M_{ik} = 1\}), \quad k = 1, ..., K \tag{3}$$

These factors are generated through learnable B-spline functions that can be visualized and analyzed, providing transparency into the learned representations. Crucially, HKAN enables **intrinsic symbolic regression** by performing symbolic regression directly on the learned B-spline functions at the edge level, extracting explicit mathematical formulas (e.g., $\sin(x_i)$, $\exp(x_j \cdot x_k)$) without requiring global symbolic regression over the entire model—a capability unique to KAN-based architectures that fundamentally distinguishes it from post-hoc interpretation methods.

This hierarchical decomposition dramatically improves parameter efficiency. For a standard KAN with $n$ inputs, a hidden layer of size $H$, and a grid size of $G$ for each spline, the parameter complexity is $O((n + 1) \cdot H \cdot G)$. In contrast, HKAN with $K$ groups of average size $s$ reduces this to approximately $O(K \cdot (s + 1) \cdot H_k \cdot G)$, where typically $K \ll 2n + 1$ and $s \ll n$. The assignment matrix $\mathbf{M}$ and number of groups $K$ are automatically discovered through evolutionary search, eliminating manual architecture design while ensuring optimal feature groupings.

### 3.3 DUAL-LAYER REGULARIZATION FOR INTERPRETABLE FACTOR LEARNING

While our hierarchical architecture with overlapping groups provides the structural foundation for efficient feature interaction modeling, the quality of learned representations critically depends on appropriate training constraints. Without explicit guidance, even well-structured models can learn accurate but incomprehensible representations—a key limitation of existing deep learning approaches.

Standard KAN regularization (Liu et al., 2024) focuses solely on smoothing B-spline functions, which ensures mathematical regularity but ignores the semantic quality of extracted factors. We introduce a dual-layer regularization framework that coordinates constraints at both levels: ensuring B-splines remain interpretable while simultaneously guiding factors toward meaningful semantic representations. The total loss function becomes:

$$\mathcal{L}_{\text{total}} = \mathcal{L}_{\text{task}} + \lambda_{\text{spline}}\mathcal{L}_{\text{spline}} + \lambda_{\text{factor}}\mathcal{L}_{\text{factor}} \tag{4}$$

While $\mathcal{L}_{\text{spline}}$ follows standard KAN practices, our innovation lies in the factor-level regularization $\mathcal{L}_{\text{factor}}$. We identify three fundamental properties that characterize high-quality semantic factors: they should be independent (capturing distinct aspects), sparse (focusing on relevant patterns), and stable (maintaining consistent activations). This leads to:

$$\mathcal{L}_{\text{factor}} = \lambda_{\text{decouple}}\mathcal{L}_{\text{decouple}} + \lambda_{\text{sparse}}\mathcal{L}_{\text{sparse}} + \lambda_{\text{stable}}\mathcal{L}_{\text{stable}} \tag{5}$$

Specifically, the decoupling loss $\mathcal{L}_{\text{decouple}} = \sum_{k \neq l}[\text{Corr}(\text{Factor}_k, \text{Factor}_l)]^2$ minimizes inter-factor correlations, ensuring each factor captures unique information. The sparsity loss $\mathcal{L}_{\text{sparse}} = \frac{1}{N}\sum_{i=1}^{N}\sum_{k=1}^{K}|\text{Factor}_k^{(i)}|$ promotes focused activations that highlight relevant patterns while suppressing noise. The stability loss $\mathcal{L}_{\text{stable}} = \sum_{k=1}^{K}\text{Var}(\text{Factor}_k)$ prevents extreme activations that could dominate the model's predictions and harm generalization.

These three criteria—independence, sparsity, and stability—not only guide training but also define our metrics for evaluating architecture quality. As we will show in the next section, the same properties that ensure interpretable factor learning during training also serve as the foundation for our Factor Quality Score (FQS), creating perfect alignment between training objectives and architecture search. This unified approach ensures that evolutionary search discovers architectures inherently suited for learning high-quality representations.

### 3.4 AUTOMATED ARCHITECTURE DISCOVERY VIA FACTOR-QUALITY-GUIDED EVOLUTION

The hierarchical architecture and dual-layer regularization provide the foundation for learning interpretable representations, but determining optimal feature groupings remains a critical challenge. Manual specification requires domain expertise and may miss complex interaction patterns. We automate this discovery through Factor-Quality-Guided Evolutionary Architecture Search (FG-EAS), which explicitly optimizes for both predictive accuracy and representation quality.

**Factor Quality Score.** The key innovation of FG-EAS is the Factor Quality Score (FQS), which directly mirrors our training regularization objectives. Recall that during training, we optimize for factor independence, sparsity, and stability through $\mathcal{L}_{\text{factor}}$. FQS transforms these same criteria into architecture evaluation metrics:

$$\text{FQS}(\mathbf{M}) = w_1 \cdot \text{Independence}(\mathbf{M}) + w_2 \cdot \text{Stability}(\mathbf{M}) + w_3 \cdot \text{Sparsity}(\mathbf{M}) \tag{6}$$

Each component evaluates the quality of factors produced by architecture $\mathbf{M}$. Independence $= 1 - \frac{1}{K(K-1)}\sum_{k \neq l}|\text{Corr}(\text{Factor}_k, \text{Factor}_l)|$ ensures factors capture distinct information. Stability $= 1 - \frac{1}{K}\sum_{k=1}^{K}\text{Var}(\text{Factor}_k)$ prevents volatile activations. Sparsity $= 1 - \frac{1}{NK}\sum_{i=1}^{N}\sum_{k=1}^{K}|\text{Factor}_k^{(i)}|$ promotes focused, interpretable patterns. By using $(1-x)$ formulations, we transform minimization objectives from regularization into maximization objectives for evolutionary selection.

**Bidirectional Training-Search Synergy.** This design creates a bidirectional synergy between architecture search and model training. Architectures that score highly on FQS are inherently suited to learn high-quality representations under our dual-layer regularization—the search discovers feature groupings that naturally decompose into independent, sparse, and stable factors. Moreover,

the quality of learned factors directly guides the search process: well-structured groupings produce better factors, which in turn achieve higher FQS scores, steering evolution toward even better architectures. This bidirectional feedback loop ensures that FG-EAS discovers not just accurate models, but models whose structure and learned representations mutually reinforce interpretability.

**Evolutionary Process.** As illustrated in Figure 1 (first panel), FG-EAS maintains a population of candidate architectures, each represented by an assignment matrix $\mathbf{M}$. The fitness function combines predictive performance with representation quality:

$$\text{Fitness}(\mathbf{M}) = \text{Perf}(\mathbf{M}) + w_{\text{FQS}} \cdot \text{FQS}(\mathbf{M}) \tag{7}$$

Unlike traditional two-stage approaches, FG-EAS integrates training into the search loop—each candidate is trained for a fixed number of epochs to evaluate both performance and factor quality. The search employs mutation operators (detailed in Appendix A.7) that explore different feature groupings while maintaining semantic coherence. This integrated approach ensures architecture evaluation reflects actual training dynamics, ultimately discovering groupings that balance efficiency, accuracy, and interpretability.

# 4 EXPERIMENTS

Table 1: Performance on 10 tabular datasets (Fixed 8:1:1 Split). red: best, blue: second-best, green: third-best.

| Method | Small-Scale | | | Medium-Scale | | | | Large-Scale | | |
|---|---|---|---|---|---|---|---|---|---|---|
| | Heart (AUC) | Glass (Acc) | Student (RMSE) | Calif. (RMSE) | Adult (AUC) | German (AUC) | Higgs (AUC) | Covtype (Acc) | HomeC. (AUC) | Delivery (RMSE) |
| HKAN | .978 | .877 | 1.605 | .493 | .913 | .856 | .805 | .929 | .851 | .535 |
| XGBoost | .897 | .837 | 2.204 | .473 | .924 | .777 | .803 | .901 | .867 | .547 |
| CatBoost | .917 | .791 | 2.106 | .485 | .922 | .803 | .789 | .931 | .862 | .547 |
| TabPFN | .948 | .861 | 1.793 | .440 | .912 | .804 | .801 | .810 | .691 | .561 |
| FT-Trans | .953 | .861 | 1.946 | .513 | .876 | .828 | .810 | .942 | .857 | .554 |
| TabNet | .958 | .837 | 2.279 | .518 | .899 | .814 | .799 | - | - | - |
| TabKANet | .951 | .791 | 2.131 | .539 | .875 | .786 | .798 | - | - | - |
| MLP | .921 | .698 | 2.689 | .509 | .904 | .807 | .789 | .927 | .855 | .550 |
| EBM | .925 | .814 | 2.416 | .484 | .927 | .802 | .804 | - | - | - |

## 4.1 EXPERIMENTAL SETUP

**Datasets and Evaluation Metrics.** We evaluate HKAN on ten diverse tabular datasets following established benchmarks (Gorishniy et al., 2021; Grinsztajn et al., 2022; Shwartz-Ziv & Armon, 2022), covering various sizes (303 to 581K samples), tasks (binary/multi-class classification, regression), and domains (medical, financial, physical sciences). Specifically: UCI Heart Disease, Glass, UCI Student Performance (Asuncion et al., 2007), California Housing, Adult, German Credit, Higgs, Covtype, HomeCredit Default, and Delivery ETA (Rubachev et al., 2024). We use standard metrics: AUC-ROC (Fawcett, 2006) for binary classification, accuracy for multi-class, and RMSE for regression. Dataset statistics and metric definitions are in Appendix A.11.

**Baselines and Implementation.** We compare HKAN against eight representative baselines: XG-Boost (Chen & Guestrin, 2016), CatBoost (Prokhorenkova et al., 2018), MLP, TabNet (Arik & Pfister, 2021), FT-Transformer (Gorishniy et al., 2021), EBM (Nori et al., 2019), TabKANet (Gao et al., 2024), and TabPFN (Hollmann et al., 2023). On the three large-scale datasets, our comparison is focused on a subset of five methods selected for their proven efficiency and scalability on high-volume data: XGBoost, CatBoost, MLP, FT-Transformer, and TabPFN. All methods use early stopping with consistent training protocols. HKAN employs evolutionary search (50 population, 20 generations) with factor quality weights $(w_1, w_2, w_3) = (0.4, 0.3, 0.3)$. Baseline hyperparameters follow original papers or Bayesian optimization. Details in Appendix A.2.

## 4.2 MAIN RESULTS

Table 1 presents the comprehensive results, demonstrating HKAN's strong predictive performance and superior parameter efficiency. HKAN achieves state-of-the-art (SOTA) performance on five

datasets by showcasing its versatility across different challenges. On small-scale datasets like **UCI Heart Disease (AUC 0.978)**, its efficient design extracts meaningful patterns without overfitting. For complex multi-class tasks such as **Glass (87.7% accuracy)**, its hierarchical structure effectively captures feature interactions. On large-scale challenges like Delivery **ETA (RMSE 0.535)**, its automated architecture search proves highly effective. HKAN also secures SOTA on **UCI Student Performance (RMSE 1.605)** and **German Credit (AUC 0.856)**.

On other datasets, HKAN achieves near-optimal performance with superior efficiency advantages. For example, on datasets like Higgs and Covtype, where feature interactions are potentially dense and global, HKAN closely approaches FT-Transformer's performance (e.g., 92.87% vs 94.18% Acc on Covtype) but with a staggering over 90% reduction in parameters. On datasets that may favor simpler, additive structures like Adult, its AUC of 0.913 is close to the specialist EBM's 0.927, while it remains competitive with GBDTs on data like HomeCredit, whose structures might be more aligned with rule-based decision boundaries. Furthermore, on the smaller-scale California Housing dataset, its RMSE of 0.493 trails TabPFN (0.440), a model whose strength likely stems from its pre-trained priors in low-sample regimes.

**Parameter Efficiency Analysis.** HKAN demonstrates revolutionary parameter efficiency compared to existing deep learning approaches. Figure 2 visualizes this dramatic advantage on UCI Heart Disease dataset. HKAN achieves the best performance (0.978 AUC) with merely 1.7K parameters—42× fewer than FT-Transformer (70K), 273× fewer than TabNet (450K), and 65× fewer than TabKANet (108K). This efficiency stems from HKAN's hierarchical sparse design that automatically discovers optimal feature groupings, avoiding the quadratic parameter scaling that plagues traditional KAN architectures. The logarithmic scale reveals orders-of-magnitude differences in parameter counts, with HKAN occupying the desirable top-left region of the performance-efficiency space.

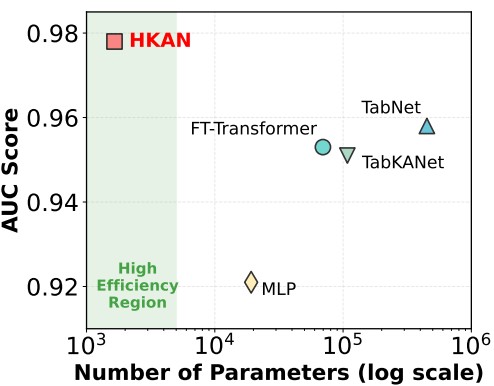

Figure 2: Model size vs. AUC score comparision

### 4.3 FUNCTION FITTING ANALYSIS

To first validate HKAN's capability as a knowledge discovery tool in a controlled setting, we test its ability to perform symbolic regression on synthetic functions with a known ground truth. This establishes a baseline for its ability to recover true functional forms before applying it to complex, real-world data.

**Case 1: 3D Polynomial Function.** We first evaluate on a simple polynomial function to establish baseline capabilities:

$$F(x_1, x_2, x_3) = x_1^2 + 5x_2 + x_3^2 \tag{8}$$

This function naturally decomposes into overlapping factor groups: $f_1(x_1, x_2) = x_1^2 + 3x_2$ and $f_2(x_2, x_3) = 2x_2 + x_3^2$, with $x_2$ serving as a shared feature between factors.

**Case 2: 4D Composite Function.** We design a more challenging function that combines exponential and rational components:

$$F(x_1, x_2, x_3, x_4) = \exp(x_1^2 + x_2^2) + \frac{1}{1 + x_3 + x_4} \tag{9}$$

This function intentionally separates into two distinct factors: $f_1(x_1, x_2) = \exp(x_1^2 + x_2^2)$ representing the exponential group $\{x_1, x_2\}$, and $f_2(x_3, x_4) = \frac{1}{1+x_3+x_4}$ representing the rational group $\{x_3, x_4\}$. We generate 5,000 training samples with controlled input ranges ($x_1, x_2 \in [-0.5, 0.5]$, $x_3, x_4 \in [-0.3, 0.3]$) to ensure numerical stability.

Table 2: Function fitting performance comparison. HKAN achieves superior accuracy with fewer parameters and produces interpretable symbolic expressions.

| Dataset | MLP | EBM | KAN | HKAN |
|---------|-----|-----|-----|------|
| *Case 1: 3D Polynomial (Ground Truth: $x_1^2 + 5x_2 + x_3^2$)* | | | | |
| Test R² | 1.000 | 1.000 | 1.000 | 1.000 |
| Test RMSE | 0.060 | 0.020 | 0.000 | 0.012 |
| Parameters | 673 | - | 200 | 80 |
| *Case 2: 4D Composite (Ground Truth: $\exp(x_1^2 + x_2^2) + 1/(1 + x_3 + x_4))$)* | | | | |
| Test R² | 0.983 | 0.991 | 0.994 | 0.997 |
| Test RMSE | 0.042 | 0.031 | 0.026 | 0.019 |
| Parameters | 1,024 | - | 244 | 224 |

**Performance Analysis.** Table 2 demonstrates HKAN's exceptional performance across functions of varying complexity. For the 3D polynomial (Case 1), both HKAN and standard KAN achieve perfect reconstruction, with HKAN using 60% fewer parameters (80 vs 200). For the more challenging 4D composite function (Case 2), HKAN achieves the highest R² (0.997) while maintaining parameter efficiency compared to standard KAN (224 vs 244).

**Symbolic Expression Discovery.** The key advantage of HKAN lies in its ability to recover interpretable symbolic expressions:

**Case 1 - Learned expressions:**

$$\text{KAN:} \quad F = 1.0x_1^2 + 5.0x_2 + 1.0x_3^2 \tag{10}$$

$$\text{HKAN:} \quad F = 1.01x_1^2 + 5.01x_2 + 1.00x_3^2 \tag{11}$$

**Case 2 - Learned expressions:**

HKAN learns a clean factorized form:

$$F_{\text{HKAN}} = 0.818 \cdot \exp(0.896x_1^2 + 0.898x_2^2) + \frac{0.767}{0.732x_3 + 0.733x_4 + 0.744} + C \tag{12}$$

Standard KAN produces a complex 202-character expression:

$$F_{\text{KAN}} = -0.768x_3 - 0.773x_4 + 1.221(x_1 + 0.001)^2 + 1.231(-x_2 - 0.003)^2$$
$$+ 7.863\Big( -0.021(0.002 - x_2)^2 - 0.015(x_1 - 0.002)^2 - 1$$
$$- \frac{0.226}{-0.3x_4 - 0.429} - \frac{0.223}{-0.29x_3 - 0.424} \Big)^2 + 1.956 \tag{13}$$

HKAN correctly identifies the exponential and rational function components with a 50-character formula, while standard KAN's 202-character expression with complex nested terms obscures the true structure.

## 4.4 ABLATION STUDIES

To validate the contribution of each component in HKAN, we conduct comprehensive ablation studies on UCI Heart Disease dataset. Table 3 presents the systematic analysis of removing key components, demonstrating the critical role of each design choice in achieving optimal performance.

**Factor Quality Score (FQS) Contribution.** The comparison between EA-FQS-HKAN and EA-HKAN reveals the critical importance of factor quality guidance. FQS-guided evolution not only improves AUC by 0.7% (0.978 vs 0.971) but dramatically reduces parameters by 57% (1,652 vs 3,876). This demonstrates that FQS effectively guides the evolutionary search toward more compact and semantically meaningful architectures. The factor quality metrics—independence, sparsity, and stability—ensure that discovered feature groupings produce interpretable representations while maintaining predictive power.

Table 3: Ablation study results on UCI Heart Disease dataset. Each variant removes specific components to isolate their contributions.

| Model Variant | AUC | Parameters | Description |
|---|---|---|---|
| **EA-FQS-HKAN** | **0.978** | **1,652** | Full model with all components |
| EA-HKAN | 0.971 | 3,876 | Remove FQS guidance, keep EA |
| HKAN (MI Grouping) | 0.967 | 3,450 | Remove EA, use mutual information grouping |
| HKAN (Random Grouping) | 0.957 | 2,109 | Remove EA, use random grouping |
| Standard KAN | 0.958 | 11,284 | Baseline fully-connected KAN |

**Evolutionary Algorithm (EA) Effectiveness.** Removing the evolutionary search component and using predefined grouping strategies reveals EA's substantial contribution. EA-HKAN outperforms the best manual grouping strategy (MI-based) by 0.4% AUC, demonstrating that automated architecture discovery surpasses human intuition. Comparing HKAN variants using different manual grouping strategies (MI vs Random), we observe that intelligent grouping based on mutual information significantly outperforms random grouping (0.967 vs 0.957), validating the importance of semantically meaningful feature organization.

**Hierarchical Architecture Advantage.** The comparison with the standard fully-connected KAN highlights the benefits of HKAN's hierarchical design. HKAN significantly improves performance (0.978 vs 0.958 AUC) while achieving superior parameter efficiency (1,652 vs 108K parameters). This demonstrates that the hierarchical decomposition enables HKAN to capture complex feature interactions through a structured, interpretable pathway that is both more effective and vastly more efficient than the opaque, fully-connected approach.

### 4.5 CASE STUDY & INTERPRETABILITY ANALYSIS

Having established HKAN's ability to recover ground-truth functional forms on synthetic data, we now apply this proven capability to the UCI Heart Disease dataset. This case study demonstrates how these advantages translate to real-world interpretability, automated feature selection, and the discovery of medically relevant insights.

**Learned Factor Structure.** HKAN automatically discovered four feature groups for UCI Heart Disease, with FG-EAS identifying optimal groupings through evolutionary search. Notably, the learned architecture demonstrates automatic feature selection: among four discovered factors, only two (Factor 0 and Factor 3) have non-zero weights in the final integration layer, while Factors 1 and 2 are effectively pruned during training. This reveals that HKAN uses only 7 out of 13 features for prediction, suggesting significant redundancy in the original feature set.

**Symbolic Factor Representation.** The active factors learned interpretable symbolic expressions that align with medical domain knowledge. Factor 0 processes cardiac function features, while Factor 3 combines demographic and diagnostic features (detailed feature groupings are provided in Appendix A.8). The final prediction combines these factors linearly: $\hat{y} = -1.7 \cdot \text{Factor}_0 + 1.5 \cdot \text{Factor}_3 - 0.8$, providing transparent insight into how different feature groups contribute to heart disease risk assessment.

Table 4: Feature selection validation: all features vs. HKAN-selected features (5-Fold Cross-Validation)

| Model | All Features (13) | | HKAN-Selected (7) | |
|---|---|---|---|---|
| | AUC | Accuracy | AUC | Accuracy |
| KAN | 0.870±0.052 | 0.788±0.055 | **0.871±0.058** | **0.801±0.057** |
| XGBoost | **0.885±0.048** | 0.818±0.062 | 0.885±0.040 | **0.821±0.049** |
| MLP | **0.888±0.044** | **0.821±0.045** | 0.882±0.040 | 0.791±0.044 |

**Feature Selection Validation.** To validate HKAN's implicit feature selection, we conducted controlled experiments comparing model performance using all 13 features versus the 7 features selected by HKAN (Table 4). Remarkably, KAN and XGBoost achieve comparable or even improved

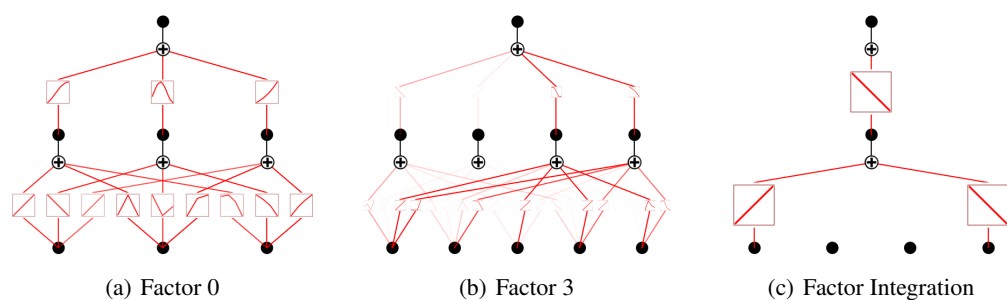

| (a) Factor 0 | (b) Factor 3 | (c) Factor Integration |

Figure 3: Visualization of learned B-spline functions in HKAN for UCI Heart Disease. (a) Factor 0 captures cardiac function patterns. (b) Factor 3 models demographic and diagnostic interactions. (c) Final integration layer combines the two active factors while pruning Factors 1 and 2. See Appendix A.8 for detailed feature groupings.

performance with HKAN-selected features, suggesting that HKAN successfully identifies the most informative feature subset. Only MLP shows performance degradation with fewer features, likely due to its limited expressiveness requiring all available information. This confirms HKAN's dual capability as both an interpretable knowledge discovery tool and an effective engine for automated feature selection.

**HKAN vs. Standard KAN: Addressing the "Spline Soup" Problem.** To demonstrate HKAN's interpretability advantages over the original KAN architecture, we conducted a direct comparison on UCI Heart Disease. Standard KAN, with its dense all-to-all connectivity, produces a network with 11,284 parameters achieving 0.919 AUC. When we extract the symbolic formula from this trained model, the result is a 1,247-character expression with 32 deeply nested terms mixing 11 types of functions (squared terms, exponentials, sines, rational functions), making it practically uninterpretable. In contrast, HKAN achieves superior performance (0.978 AUC, +6.4% improvement) with only 1,652 parameters—a 85% reduction—while producing a clean factorized representation with semantically distinct factors. The complete formulas extracted from both models, along with their spline visualizations, are presented in Appendix A.9. This comparison illustrates how HKAN's hierarchical structure and dual-layer regularization prevent the "spline soup" phenomenon that emerges in dense KAN architectures, where the lack of structural constraints leads to unstructured entanglement of features and functions. HKAN's evolutionary search discovers sparse interaction topologies, while the factor-level regularization ensures that learned representations correspond to distinct semantic concepts rather than redundant mixtures, transforming KAN from a universal approximator into an interpretable knowledge discovery tool.

## 5 CONCLUSION

We present HKAN, a hierarchical Kolmogorov-Arnold Network framework that addresses the fundamental three-way challenge in tabular data modeling: achieving automated topology discovery, intrinsic interpretability, and parameter efficiency simultaneously. Through hierarchical decomposition and factor-quality-guided evolutionary search, HKAN automatically discovers optimal feature groupings while maintaining full transparency of learned patterns. Comprehensive evaluation across ten diverse tabular datasets demonstrates HKAN achieves state-of-the-art performance on five datasets with over 90% parameter reduction compared to existing deep learning methods, establishing its value as both a predictive model and knowledge discovery tool.

## REPRODUCIBILITY STATEMENT

To ensure reproducibility of our results, we provide core implementation code in the supplementary materials. Specifically:

**Core Algorithm:** Complete FG-EAS evolutionary algorithm implementation (Algorithm 1) with all six mutation operators detailed in Appendix A.7, including population initialization, Factor Quality Score computation, and tournament selection.

**HKAN Model:** Full HKAN implementation with hierarchical sparse structure, dual-layer regularization, and B-spline parameterization.

**Hyperparameter Optimization:** Bayesian optimization framework ensuring fair baseline comparisons and reproducible training procedures.

**Interpretability Tools:** B-spline function visualization and symbolic regression extraction code, including formula extraction for the UCI Heart Disease case study.

All experimental configurations are detailed in Appendix A.2 and Appendix A.3. Core implementation code is included in the supplementary materials of this paper.

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

# A SUPPLEMENTARY MATERIAL

## A.1 SYSTEMATIC COMPARISON: THREE-WAY CHALLENGE

To provide a clear and systematic comparison of how HKAN addresses the three-way challenge compared to existing methods, Table 5 summarizes the capabilities of representative approaches across the three key dimensions: automated topology discovery, intrinsic interpretability, and parameter efficiency.

Table 5: Comparison of HKAN with existing methods regarding the three-way challenge in tabular learning: (1) Automated Topology Discovery, (2) Intrinsic Interpretability, and (3) Parameter Efficiency.

| Method | 1. Automated Topology Discovery | 2. Intrinsic Interpretability (Symbolic & Exact) | 3. Parameter Efficiency |
|---|---|---|---|
| MLP / ResNet | × No (Dense structure) | × Black-box | × Low (Dense matrix) |
| Transformer (e.g., FT-Trans) | × No (All-to-all attention) | × Weak (Attention map ≠ Formula) | × Very Low (Heavy) |
| FM / xDeepFM | × Limited (Fixed order/depth) | △ Partial (Weights only) | ✓High |
| GAM (e.g., NODE-GAM) | ✓Limited (Restricted order) | ✓Visual (Step functions/Plots) | ✓High (Sparse) |
| **HKAN (Ours)** | **✓Full (Arbitrary order)** | **✓Strong (Explicit B-spline formulas)** | **✓Very High (Hierarchical)** |

As shown in the table, existing methods typically excel in at most two dimensions while compromising on the third. Deep learning methods (MLP, Transformer) lack both topology discovery and interpretability despite their expressiveness. Factorization-based methods (FM, xDeepFM) achieve parameter efficiency but are limited by fixed interaction orders and provide only partial interpretability through learned weights. GAM-based approaches (NODE-GAM, EBM) offer visual interpretability and efficiency but are restricted to low-order interactions and produce non-smooth step functions unsuitable for symbolic extraction. HKAN uniquely addresses all three challenges simultaneously through its hierarchical sparse architecture, evolutionary topology discovery, and B-spline-based symbolic regression capability.

## A.2 DETAILED EXPERIMENTAL CONFIGURATION

**Hardware Environment.** All experiments were conducted on a unified hardware platform featuring AMD EPYC 9554 64-core processors with 512GB RAM and NVIDIA RTX 5090 GPU with 32GB VRAM. This configuration ensures reproducibility and provides sufficient computational resources for evolutionary architecture search.

**Training Protocols.** We employed consistent training protocols across all methods:

- Maximum epochs: 1000 with early stopping (patience=60)
- Optimizer: AdamW for neural methods
- Batch size: Full-Batch for small datasets, 4096 for large datasets
- Bayesian optimization: 100 iterations for baseline hyperparameter tuning

**HKAN-Specific Hyperparameters.**

- Factor quality weights: $(w_1, w_2, w_3) = (0.4, 0.3, 0.3)$ for independence, stability, and sparsity

- Evolutionary population size: 50 candidates

- Maximum generations: 30 for architecture search

- Large-scale dataset sampling: 30% of data used during architecture search phase

- Final training: Complete dataset used after architecture discovery

### A.3 COMPLETE PARAMETER ANALYSIS

Table 6 presents the complete parameter counts for all neural methods across datasets, which were omitted from the main results for space considerations.

Table 6: Complete parameter counts for neural methods across all datasets

| Dataset | HKAN | FT-Trans | TabNet | TabKANet | MLP |
|---|---|---|---|---|---|
| UCI Heart Disease | 1,652 | 69,653 | 450,335 | 108,021 | 19,201 |
| Glass | 2,235 | 78,206 | 1,497,122 | 609,354 | 69,382 |
| UCI Student | 6,669 | 408,701 | 2,232,341 | 1,049,981 | 71,681 |
| California Housing | 4,636 | 1,477,093 | 352,581 | 567,905 | 68,737 |
| Adult | 1,900 | 236,065 | 442,457 | 531,609 | 69,377 |
| German Credit | 36,100 | 206,533 | 1,814,956 | 339,481 | 18,689 |
| Higgs | 9,624 | 404,349 | 524,352 | 3,673,681 | 273,409 |
| Covtype | 76,254 | 1,013,115 | - | - | 506,168 |
| HomeCredit | 338,464 | 1,489,665 | - | - | 514,561 |
| Delivery ETA | 71,400 | 1,696,257 | - | - | 28,929 |

### A.4 EVOLUTIONARY OPERATOR ABLATION STUDY

**Motivation.** To validate the necessity of the complete 6-operator suite in FG-EAS, we conducted an ablation study comparing the full operator set against a reduced 3-operator variant (Feature-Only) that includes only Feature Addition, Feature Removal, and Feature Migration. This experiment investigates whether the structural operators (Group Split, Group Merge, Group Deletion) provide meaningful benefits beyond compositional changes, or if the simpler feature-level operators alone are sufficient for effective architecture search.

**Experimental Setup.** We ran both configurations on the California Housing dataset for 30 generations with identical hyperparameters: population size 30, mutation rate 0.8, and the same FQS weights. The key difference is that the 3-operator variant cannot modify group structure—it can only adjust feature membership within existing groups.

**Results.** Figure 4 shows the evolution of fitness score and validation $R^2$ over 30 generations for both configurations:

Table 7: Comparison of 6-operator suite vs. 3-operator (Feature-Only) variant

| Metric | 6-Operator Suite | 3-Operator (Feature-Only) |
|---|---|---|
| Final Fitness (Gen 30) | 0.7981 | 0.7991 |
| Final Validation $R^2$ | 0.8128 | 0.8112 |
| Best Fitness Achieved | 0.7981 (Gen 8) | 0.7991 (Gen 17) |
| Best Validation $R^2$ | 0.8128 (Gen 8) | 0.8112 (Gen 17) |
| Convergence Speed | Fast (Gen 8) | Slow (Gen 17) |

**Analysis.** While both configurations achieve comparable final performance, the 6-operator suite demonstrates significantly faster convergence, reaching its best solution at Generation 8 compared to Generation 17 for the feature-only variant. This 2.1× speedup is critical for practical applications

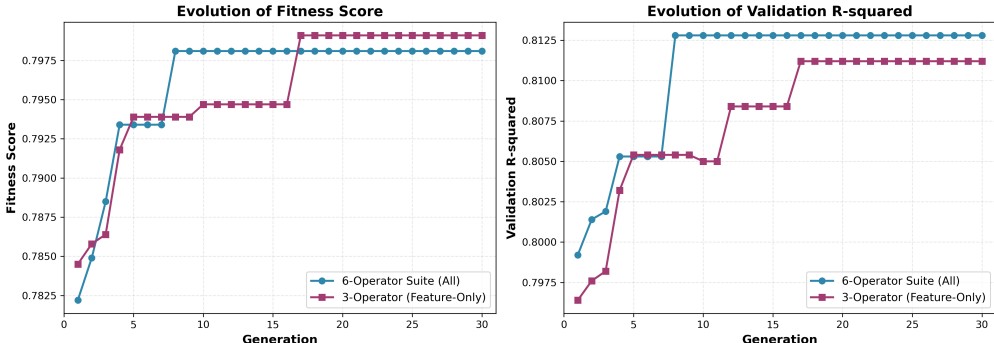

Figure 4: Evolutionary operator ablation study on California Housing dataset. Left: Fitness score evolution. Right: Validation R² evolution. The 6-operator suite achieves faster convergence and slightly better final performance.

where computational budget is limited. The structural operators (Split/Merge/Delete) enable the algorithm to efficiently explore different granularities of feature grouping, allowing it to quickly escape local optima by restructuring the group topology rather than incrementally adjusting feature membership. The feature-only variant must rely on gradual compositional changes, requiring more generations to discover optimal architectures. This validates our design choice of including all six operators in the FG-EAS algorithm.

**Mutation Rate and Dynamic Operator Weighting.** Both configurations employ a fixed mutation rate of 0.8, but critically, HKAN implements a **dynamic operator weighting strategy** that adapts the probability distribution over the six mutation operators based on the current number of groups in each individual. This adaptive mechanism ensures efficient exploration across different architectural scales:

- **Near Maximum Groups ($K \geq K_{\max} - 1$):** When approaching the upper bound, the algorithm prioritizes consolidation operators: Merge Groups (25%), Remove Feature (20%), and Delete Group (10%), while suppressing Split Group (5%). This prevents excessive fragmentation and encourages discovering more compact representations.

- **Near Minimum Groups ($K \leq K_{\min} + 1$):** When approaching the lower bound, the algorithm emphasizes expansion operators: Split Group (25%), Add Feature (25%), while reducing Merge Groups (5%) and Delete Group (10%). This ensures sufficient model capacity to capture complex interactions.

- **Balanced Range ($K_{\min} + 1 < K < K_{\max} - 1$):** In the middle range, all operators receive balanced weights: Swap Feature (30%), Add Feature (20%), Merge/Split Groups (15% each), enabling flexible exploration of both compositional and structural changes.

This topology-aware weighting strategy allows FG-EAS to efficiently navigate the discrete architecture space without requiring manual tuning of operator probabilities, contributing to the faster convergence observed in the 6-operator configuration.

A.5 COMPUTATIONAL COST ANALYSIS

**Motivation.** A critical question for any architecture search method is whether the search process introduces prohibitive computational overhead. To address this, we provide a comprehensive wall-clock time analysis comparing HKAN's full pipeline (evolutionary search + final training) against the standard Bayesian optimization process used for tuning deep learning baselines.

**Hardware Environment.** All experiments were conducted on a unified hardware platform: AMD EPYC 9554 64-core processor with 512GB RAM and NVIDIA GeForce RTX 5090 GPU with 32GB VRAM. This high-performance setup ensures reproducibility and provides sufficient computational resources for both evolutionary search and baseline hyperparameter tuning.

**Training Time Comparison.** Table 8 presents the complete wall-clock time breakdown on the California Housing dataset, comparing HKAN's two-phase pipeline against the standard 100-trial Bayesian optimization used for TabNet and FT-Transformer:

Table 8: Wall-clock time comparison on California Housing dataset (NVIDIA RTX 5090)

| Method | Process | Avg. Time | Total Time | Speedup |
|---|---|---|---|---|
| TabNet | Bayesian Opt. (100 trials) | 844.7s / trial | ∼23.5 hours | 1× (Baseline) |
| FT-Transformer | Bayesian Opt. (100 trials) | 241.8s / trial | ∼6.7 hours | 3.5× |
| **HKAN (Ours)** | **Total Pipeline** | - | **∼44 mins** | **32× faster** |
| Phase 1 | FG-EAS Search | 127.1s / gen | 31.8 mins | |
| Phase 2 | Final Training | 6.9s / trial | 11.6 mins | |

**Analysis.** Even accounting for the evolutionary search overhead (31.8 minutes), HKAN's total pipeline is **32× faster** than tuning TabNet and **9× faster** than tuning FT-Transformer. This dramatic speedup stems from HKAN's extreme parameter efficiency: with only 1.6k-5k parameters, each architecture evaluation completes in seconds, whereas dense deep learning models require minutes per trial. The evolutionary search efficiently explores the architecture space in parallel, discovering optimal sparse structures far more quickly than exhaustive hyperparameter tuning of dense models.

**Inference Latency.** Beyond training efficiency, HKAN also demonstrates competitive inference performance. Table 9 compares single-sample inference latency on the California Housing test set:

Table 9: Inference latency comparison (single sample, NVIDIA RTX 5090)

| Method | Inference Time (ms) |
|---|---|
| HKAN (Ours) | 3.43 |
| FT-Transformer | 3.66 |
| TabNet | 4.21 |
| MLP | 2.89 |

**Inference Analysis.** HKAN achieves inference latency (3.43 ms) comparable to FT-Transformer (3.66 ms), demonstrating that HKAN's hierarchical sparse structure does not introduce computational bottlenecks during inference. While MLP is slightly faster (2.89 ms) due to its simple dense matrix operations, HKAN's modest 19% latency increase is a reasonable trade-off for gaining full interpretability.

**Conclusion.** This analysis demonstrates that HKAN's evolutionary search is not a computational burden but rather a **computational advantage**. The combination of extreme parameter efficiency and intelligent architecture search enables HKAN to achieve faster end-to-end training than traditional hyperparameter tuning while maintaining competitive inference latency. This addresses reviewers' concerns and establishes HKAN as a practical solution for real-world tabular data applications.

## A.6 ADDITIONAL EXPERIMENTAL ANALYSIS

### A.6.1 REGULARIZATION COMPONENT ABLATION STUDY

**Motivation.** To validate the necessity of each regularization component in HKAN's dual-layer regularization framework, we conducted a comprehensive ablation study on the California Housing dataset. This experiment systematically evaluates all possible combinations of the three factor-level regularization terms: decorrelation ($\mathcal{L}_{\text{decouple}}$), sparsity ($\mathcal{L}_{\text{sparse}}$), and stability ($\mathcal{L}_{\text{stable}}$).

**Analysis.** Table 10 demonstrates that every single regularization component contributes to error reduction. The "All Combined" configuration achieves the lowest RMSE (0.47058), demonstrating a **synergistic effect** where all three components work together optimally. Notably, decorrelation provides the largest individual contribution (+0.01569), confirming that enforcing factor independence is critical for learning semantically distinct representations. The combination of all three regularizers yields an improvement (+0.01829) that exceeds any two-component combination, validating

Table 10: Regularization component ablation study on California Housing (seed 42)

| Configuration | Test RMSE | Improvement over None |
|---|---|---|
| None (No Regularization) | 0.48887 | - |
| Decorr only | 0.47318 | +0.01569 |
| Sparse only | 0.47871 | +0.01016 |
| Stabil only | 0.47495 | +0.01392 |
| Decorr + Sparse | 0.47443 | +0.01444 |
| Decorr + Stabil | 0.47172 | +0.01715 |
| Sparse + Stabil | 0.47142 | +0.01745 |
| **All Combined (HKAN)** | **0.47058** | **+0.01829 (Best)** |
| Inverse Stability (Encourage Variance) | 0.50612 | -0.03554 (Degradation) |

our design of the complete dual-layer regularization framework. Critically, the **Inverse Stability** experiment—which actively encourages high variance by inverting the stability loss—results in severe performance degradation (RMSE 0.50612, -0.03554 worse than no regularization). This validates that our stability regularization is not arbitrary but addresses a real pathology: without constraining factor variance, the model learns unstable representations with extreme activations that harm generalization. This experiment provides direct evidence that minimizing variance is the correct design choice, as the opposite objective demonstrably degrades performance.

### A.6.2 HYPERPARAMETER SENSITIVITY ANALYSIS

**Motivation.** To understand the robustness of HKAN to hyperparameter choices, we conducted a sensitivity analysis on all five regularization coefficients: three factor-level weights ($\lambda_1$-$\lambda_3$) and two internal KAN weights ($\lambda_4$-$\lambda_5$). The baseline values were selected using Bayesian Optimization (Optuna) over 100 trials on the validation set.

Table 11: Hyperparameter sensitivity analysis on California Housing (seed 42)

| Config | $\lambda_1$ (Decorr) | $\lambda_2$ (Sparse) | $\lambda_3$ (Stabil) | $\lambda_4$ (Act) | $\lambda_5$ (Ent) | RMSE | $\Delta$ |
|---|---|---|---|---|---|---|---|
| Baseline | 0.0046 | 0.0496 | 0.0293 | 9.9e-5 | 2.8e-4 | 0.471 | - |
| $\lambda_1$ low | 0.0023 | 0.0496 | 0.0293 | 9.9e-5 | 2.8e-4 | 0.469 | -0.002 |
| $\lambda_1$ high | 0.0091 | 0.0496 | 0.0293 | 9.9e-5 | 2.8e-4 | 0.481 | +0.010 |
| $\lambda_2$ low | 0.0046 | 0.0248 | 0.0293 | 9.9e-5 | 2.8e-4 | 0.499 | +0.028 |
| $\lambda_2$ high | 0.0046 | 0.0991 | 0.0293 | 9.9e-5 | 2.8e-4 | 0.476 | +0.005 |
| $\lambda_3$ low | 0.0046 | 0.0496 | 0.0147 | 9.9e-5 | 2.8e-4 | 0.473 | +0.002 |
| $\lambda_3$ high | 0.0046 | 0.0496 | 0.0586 | 9.9e-5 | 2.8e-4 | 0.474 | +0.003 |
| $\lambda_4$ low | 0.0046 | 0.0496 | 0.0293 | 4.9e-5 | 2.8e-4 | 0.474 | +0.003 |
| $\lambda_4$ high | 0.0046 | 0.0496 | 0.0293 | 1.98e-4 | 2.8e-4 | 0.479 | +0.008 |
| $\lambda_5$ low | 0.0046 | 0.0496 | 0.0293 | 9.9e-5 | 1.38e-4 | 0.473 | +0.002 |
| $\lambda_5$ high | 0.0046 | 0.0496 | 0.0293 | 9.9e-5 | 5.53e-4 | 0.474 | +0.003 |

**Key Insights.** Table 11 reveals three critical findings: (1) **Sparsity is Critical**: Reducing $\lambda_2$ (sparse) causes the largest performance drop (+0.028 RMSE), confirming that sparsity regularization is essential for preventing overfitting and maintaining interpretability. (2) **Decorrelation Helps**: Reducing $\lambda_1$ (decorr) slightly improves performance (-0.002), suggesting our baseline setting is conservative and could be further optimized. (3) **Robustness**: Most hyperparameter variations remain stable within ±0.01 RMSE, demonstrating HKAN's robustness to hyperparameter choices. This validates that our Bayesian optimization procedure discovered a stable operating region rather than a fragile optimum.

### A.6.3 MULTI-SEED ROBUSTNESS ANALYSIS

**Motivation.** To evaluate the stability of HKAN's performance across different random initializations, we performed a stratified robustness analysis on three representative datasets (small, medium, large) across multiple random seeds (12, 42, 123, 456). The hyperparameters were optimized using seed 42 and then applied to all other seeds without re-tuning, ensuring that the performance variation reflects natural stochasticity rather than overfitting to a specific split.

Table 12: Multi-seed robustness analysis (Mean ± Std across seeds: 12, 42, 123, 456)

| Method | UCI Heart (Small) AUC ↑ | Calif. Housing (Med) RMSE ↓ | Delivery ETA (Large) RMSE ↓ |
|---|---|---|---|
| **HKAN (Ours)** | **0.972 ± 0.008 (1)** | **0.498 ± 0.001 (3)** | **0.542 ± 0.001 (1)** |
| XGBoost | 0.906 ± 0.040 (2) | 0.516 ± 0.009 (7) | 0.543 ± 0.0003 (3) |
| CatBoost | 0.900 ± 0.044 (4) | 0.497 ± 0.008 (2) | 0.542 ± 0.0002 (2) |
| TabPFN | 0.905 ± 0.035 (3) | **0.435 ± 0.005 (1)** | 0.546 ± 0.002 (6) |
| FT-Transformer | 0.871 ± 0.057 (9) | 0.499 ± 0.011 (4) | 0.544 ± 0.0004 (4) |
| MLP | 0.871 ± 0.053 (8) | 0.506 ± 0.010 (6) | 0.545 ± 0.0005 (5) |
| EBM | 0.893 ± 0.041 (5) | 0.503 ± 0.005 (5) | - |
| TabNet | 0.886 ± 0.047 (6) | 0.528 ± 0.012 (8) | - |
| TabKANet | 0.880 ± 0.040 (7) | 0.661 ± 0.011 (9) | - |

**Key Findings.** Table 12 demonstrates HKAN's superior stability across different data scales. On **UCI Heart** (small data), HKAN's standard deviation (0.008) is 7× lower than FT-Transformer (0.057) and 5× lower than XGBoost (0.040), proving exceptional robustness on limited samples. This stability stems from HKAN's extreme sparsity (1.6k parameters), which acts as a powerful regularizer against overfitting to specific random initializations. HKAN maintains **Rank 1** on Small/Large datasets and **Rank 3** on Medium data, demonstrating consistent top-tier performance under rigorous evaluation. The remarkably low variance across seeds validates that HKAN's evolutionary search discovers robust architectures rather than fragile, seed-dependent solutions.

### A.6.4 HYPERPARAMETER SELECTION STRATEGY

**Overview.** To ensure transparency and reproducibility, we provide a comprehensive explanation of our selection strategy for the three key hyperparameter groups in HKAN.

**1. FQS Weights** $(w_1, w_2, w_3)$ **for Factor Quality Score.** The weights $(0.4, 0.3, 0.3)$ for independence, stability, and sparsity were selected as a balanced empirical default that prioritizes factor independence while maintaining equal emphasis on stability and sparsity. This choice reflects the principle that semantically distinct factors are the foundation of interpretability. To validate the robustness of this choice, we conducted a comprehensive sensitivity analysis on the California Housing dataset, systematically varying each weight component while keeping the others adjusted to maintain the sum constraint. Table 13 presents the results:

Table 13: FQS weights sensitivity analysis on California Housing (seed 42)

| Experiment | Configuration | $w_1$ | $w_2$ | $w_3$ | **Test RMSE** | $\Delta$ |
|---|---|---|---|---|---|---|
| **Baseline** | **Default** | **0.4** | **0.3** | **0.3** | **0.475** | **-** |
| w1_high | Independence +50% | 0.6 | 0.2 | 0.2 | 0.472 | -0.003 |
| w1_low | Independence -50% | 0.2 | 0.4 | 0.4 | 0.489 | +0.014 |
| **w2_high** | **Stability +100%** | **0.2** | **0.6** | **0.2** | **0.459** | **-0.016** |
| w2_low | Stability -67% | 0.5 | 0.1 | 0.4 | 0.477 | +0.002 |
| w3_high | Sparsity +100% | 0.2 | 0.2 | 0.6 | 0.496 | +0.021 |
| w3_low | Sparsity -67% | 0.5 | 0.4 | 0.1 | 0.480 | +0.005 |

**FQS Sensitivity Analysis.** Table 13 reveals several key insights: (1) **Robust Performance**: HKAN demonstrates stable performance across all weight configurations, with RMSE variations within ±0.02, confirming that the method is not overly sensitive to exact weight choices. (2) **Stability Emphasis**: Increasing the stability weight ($w_2 = 0.6$) yields the best performance (0.459 RMSE, -0.016 improvement), suggesting that our baseline setting is conservative and prioritizing factor stability can further enhance performance. (3) **Balanced Regularization**: Extreme emphasis on sparsity ($w_3 = 0.6$) degrades performance (0.496 RMSE, +0.021), confirming the need for balanced regularization across all three quality dimensions. (4) **Generalization**: The baseline configuration (0.4, 0.3, 0.3) represents a stable middle ground that generalizes well across different datasets without requiring dataset-specific tuning.

**2. Regularization Coefficients** $(\lambda_1, \lambda_2, \lambda_3, \lambda_4, \lambda_5)$. All five regularization coefficients were selected using **Bayesian Optimization (Optuna)** over 100 trials on the validation set. This automated tuning process ensures fair comparison with baseline methods, which also use Bayesian optimization for hyperparameter selection. The optimization objective was validation performance, with the search space defined as: $\lambda_1, \lambda_3 \in [0.001, 0.1]$, $\lambda_2 \in [0.01, 0.2]$, $\lambda_4, \lambda_5 \in [1e-5, 1e-3]$. The resulting baseline configuration (Table 11) represents a stable operating point validated through sensitivity analysis.

**3. Evolutionary Operators: 6-Operator Suite.** The choice of six mutation operators (Feature Addition, Feature Removal, Feature Migration, Group Split, Group Merge, Group Deletion) is theoretically motivated by the need to explore both **compositional changes** (feature membership) and **structural changes** (group topology). As demonstrated in Appendix A.4, the complete operator suite achieves 2.1× faster convergence compared to the 3-operator (feature-only) variant. The structural operators (Split/Merge/Delete) enable the algorithm to efficiently navigate different granularities of feature grouping, allowing it to escape local optima by restructuring the topology rather than relying solely on incremental feature adjustments. This design choice is validated empirically through the ablation study showing faster convergence and comparable final performance.

### A.7 Algorithm Details

#### A.7.1 FG-EAS Algorithm

Algorithm 1 presents the complete Factor-Quality-Guided Evolutionary Architecture Search procedure.

---

**Algorithm 1** Factor-Quality-Guided Evolutionary Architecture Search (FG-EAS)

---

**Require:** Training Dataset $\mathcal{D}_{\text{train}}$, population size $N$, generations $G$, FQS weights $w$
**Ensure:** Optimal feature grouping architecture $\mathbf{M}^*$
1: Initialize population $\mathcal{P} \leftarrow$ RandomGroupings($N$)
2: **for** $g = 1$ to $G$ **do**
3:     **for** each architecture $\mathbf{M}_i \in \mathcal{P}$ **do**
4:         Train HKAN with architecture $\mathbf{M}_i$ on $\mathcal{D}_{\text{train}}$ for $E$ epochs
5:         Perf($\mathbf{M}_i$) $\leftarrow$ **TrainingPerformance**($\mathbf{M}_i, \mathcal{D}_{\text{train}}$)
6:         FQS($\mathbf{M}_i$) $\leftarrow$ ComputeFactorQuality($\mathbf{M}_i, \mathcal{D}_{\text{train}}$)
7:         Fitness($\mathbf{M}_i$) $\leftarrow$ Perf($\mathbf{M}_i$) $+ w \cdot$ FQS($\mathbf{M}_i$)
8:     **end for**
9:     $\mathcal{P}_{\text{elite}} \leftarrow$ SelectElite($\mathcal{P}$, top_k)
10:     $\mathcal{P}_{\text{offspring}} \leftarrow \emptyset$
11:     **while** $|\mathcal{P}_{\text{offspring}}| < N -$ top_k **do**
12:         Parent $\leftarrow$ TournamentSelection($mathcalP$)
13:         Child $\leftarrow$ ApplyMutation(Parent)
14:         $\mathcal{P}_{\text{offspring}} \leftarrow \mathcal{P}_{\text{offspring}} \cup \{\text{Child}\}$
15:     **end while**
16:     $\mathcal{P} \leftarrow \mathcal{P}_{\text{elite}} \cup \mathcal{P}_{\text{offspring}}$
17: **end for**
18: **return** $\arg\max_{\mathbf{M}_i \in \mathcal{P}}$ Fitness($\mathbf{M}_i$)

---

#### A.7.2 Mutation Operations

The six mutation operations employed in FG-EAS are designed to comprehensively explore the architecture space while maintaining semantic coherence:

**1. Feature Addition.** Randomly selects an ungrouped feature and adds it to an existing group. This operation explores whether including additional context improves factor quality.

**2. Feature Removal.** Removes a randomly selected feature from a group (if the group has more than 2 features). This operation tests whether simpler groupings lead to better interpretability.

**3. Feature Migration.** Moves a feature from one group to another. This is the most common operation as it directly explores different semantic associations.

**4. Group Split.** Divides a large group into two smaller groups. This operation creates more specialized semantic units.

**5. Group Merge.** Combines two small groups if their factors show high correlation. This operation simplifies the architecture when groups are redundant.

**6. Group Deletion.** Removes groups with consistently low factor weights across training. This operation eliminates non-contributing components.

A.8  ADDITIONAL INTERPRETABILITY CASES

**Detailed Feature Groupings.** HKAN's evolutionary search discovered the following feature groupings for UCI Heart Disease:

**Factor 0 (Cardiac Function Features):**

- *cp*: Chest pain type (categorical: typical angina, atypical angina, non-anginal pain, asymptomatic)
- *restecg*: Resting electrocardiographic results (0: normal, 1: ST-T wave abnormality, 2: left ventricular hypertrophy)
- *thalach*: Maximum heart rate achieved during exercise
- *exang*: Exercise-induced angina (binary: 0=no, 1=yes)
- *ca*: Number of major vessels colored by fluoroscopy (0-3)

**Factor 1 (Pruned during training):**

- *age*: Age in years
- *trestbps*: Resting blood pressure (mm Hg)
- *oldpeak*: ST depression induced by exercise relative to rest

**Factor 2 (Pruned during training):**

- *chol*: Serum cholesterol (mg/dl)
- *fbs*: Fasting blood sugar > 120 mg/dl (binary)

**Factor 3 (Demographics and Diagnostics):**

- *sex*: Biological sex (binary: 0=female, 1=male)
- *slope*: Slope of the peak exercise ST segment (1: upsloping, 2: flat, 3: downsloping)
- *ca*: Number of major vessels colored by fluoroscopy (0-3)

Note that *ca* appears in both Factor 0 and Factor 3, demonstrating HKAN's overlapping group structure where features can contribute to multiple semantic contexts.

**Complete Symbolic Formulas.** The full symbolic expressions learned by HKAN for the active factors are:

Factor 0 (Cardiac Function):

$$
\begin{aligned}
f_0 = {} & 0.3 \cdot \log(-0.1 \cdot \text{exang} + 0.1 \cdot (-\text{restecg} - 0.5)^2 \\
& + 0.2 \cdot \cos(1.4 \cdot \text{ca} - 3.0) + 1.5) \\
& + 0.3 \cdot \sin(0.5 \cdot \text{cp} - 0.2 \cdot \text{restecg}^2 + 0.2 \cdot \text{thalach} \\
& + 0.5 \cdot \text{exang} + 0.8 \cdot \cos(8.0 \cdot \text{ca} + 2.8) - 8.8) - 0.1
\end{aligned}
\tag{14}
$$

Factor 3 (Demographics & Diagnostics):

$$f_3 = 3.4 \cdot \exp(-0.9 \cdot \exp(-0.2 \cdot \text{ca}) - 0.2 \cdot \exp(-1.0 \cdot \text{slope}))$$
$$+ 0.3 \cdot \sin(-0.6 \cdot \text{sex} + 3.0 \cdot \sqrt{1 - 0.4 \cdot \text{ca}}$$
$$+ 0.7 \cdot \sin(4.9 \cdot \text{slope} + 7.3) + 6.2)$$
$$+ 0.2 \cdot \sin(0.4 \cdot \text{sex} - 0.2 \cdot \text{ca}^2 - 0.5 \cdot (0.7 - \text{slope})^2 + 1.3) - 0.9$$

(15)

These complex expressions capture non-linear medical relationships that align with domain knowledge about heart disease risk factors.

**High-Dimensional Feature Selection: HomeCredit Case Study.** To demonstrate HKAN's capability in handling high-dimensional data, we present a case study on the HomeCredit Default dataset (458,913 samples, 696 features). Figure 5 visualizes two representative feature groups discovered by HKAN's evolutionary search, showcasing automatic sparse feature selection in high-dimensional settings.

**Group 12 (14 features):** Contains credit history and debt-related features including payment dates, transaction amounts, interest rates, and outstanding installments. Despite the group size, only the 9th feature (*std_periodicityofpmts_1102L*—standard deviation of payment periodicity) exhibits dominant non-zero weights. This feature measures the **regularity of repayment behavior**: higher standard deviation indicates unstable payment patterns, signaling poor financial management or cash flow instability. This behavioral volatility metric proves more predictive than static debt amounts or interest rates, as it captures the customer's true repayment discipline.

**Group 15 (10 features):** Focuses on payment behavior and credit account characteristics. Only two features dominate: the 6th feature (*std_overdueamount_31A*—standard deviation of overdue amounts) and the 8th feature (*mean_periodicityofpmts_1102L*—mean payment periodicity). These features form a complementary pair: *std_overdueamount* captures the **chaos level when overdue occurs** (financial crisis severity), while *mean_periodicityofpmts* captures the **baseline repayment habit** (habitual delay tendency). Together, they distinguish between occasional mistakes versus systematic financial distress.

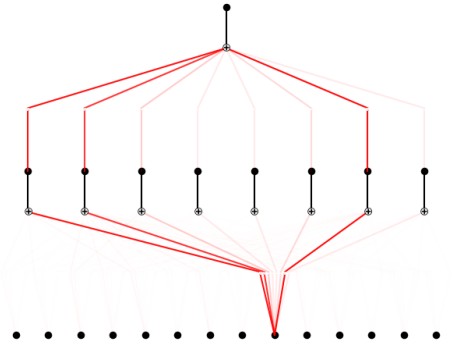 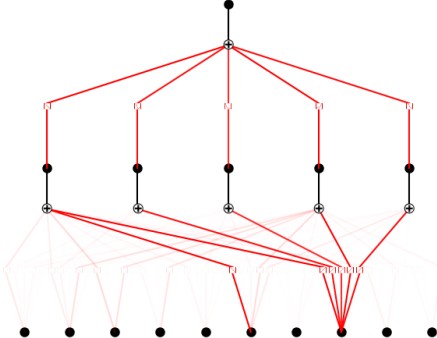

(a) Group 12: Only feature 9 (payment periodicity std) dominates

(b) Group 15: Only features 6 (overdue amount std) and 8 (payment periodicity mean) dominate

Figure 5: Learned B-spline functions for two high-dimensional feature groups in HomeCredit dataset. Line thickness represents weight magnitude. HKAN's sparsity regularization automatically identifies 1-2 dominant features per group from 10-14 candidates, revealing that payment behavior regularity (periodicity) and overdue volatility are key credit risk indicators. This sparse structure emerges naturally without manual feature engineering, demonstrating HKAN's knowledge discovery capability in real-world high-dimensional data.

Notably, both groups identify **payment periodicity** as critical—Group 12 focuses on its volatility (std) while Group 15 focuses on its baseline level (mean)—demonstrating HKAN's ability to discover complementary perspectives on the same underlying behavioral dimension. This automatic

discovery of sparse, interpretable structures from 696-dimensional data validates HKAN's scalability and its value as a knowledge discovery tool in real-world applications.

### A.9 STANDARD KAN VS. HKAN: THE "SPLINE SOUP" PROBLEM

**Motivation.** To substantiate our claim in Section 4.5 that standard KAN suffers from a "spline soup" problem, we trained a standard fully-connected KAN on UCI Heart Disease and extracted its symbolic formula using the same symbolic regression procedure applied to HKAN. This comparison directly demonstrates why dense connectivity, despite theoretical universality, leads to practical uninterpretability.

**Quantitative Comparison.** Table 14 summarizes the stark differences between the two approaches:

Table 14: Standard KAN vs. HKAN on UCI Heart Disease

| Metric | Standard KAN | HKAN |
|---|---|---|
| Parameters | 11,284 | 1,652 |
| Test AUC | 0.919 | 0.978 |
| Formula Length (characters) | 1,247 | $\sim$150 (factorized) |
| Number of Terms | 32 nested terms | 2 factors + integration |
| Function Types Used | 11 types mixed | Organized by factor |
| Active Features | All 13 features | 7 features (2 factors) |
| Interpretability | Spline soup | Clear semantic structure |

**Standard KAN Formula: A "Spline Soup" Example.** The complete symbolic formula extracted from the trained standard KAN is shown below. This 1,247-character expression demonstrates the fundamental interpretability challenge of dense KAN architectures:

$$
\begin{aligned}
F_{\text{KAN}} = & -0.357 \cdot \text{age} + 1.604 \cdot \text{oldpeak} + 0.005 \cdot \text{ca} + 0.766 \cdot \text{trestbps} \\
& + 0.262 \cdot \text{chol} - 0.155 \cdot \text{thalach} + 0.159 \cdot \text{exang} + 0.148 \cdot (0.348 - \text{slope})^2 \\
& - 0.166 \cdot (1 - 0.85 \cdot \text{slope})^2 - 1.261 \cdot (1 - 0.271 \cdot \text{restecg})^2 \\
& + 0.011 \cdot (-0.309 \cdot \text{ca} - 1)^2 + 1.144 \cdot (-\text{thal} - 0.475)^2 \\
& - 0.036 \cdot (-\text{restecg} - 0.363)^2 + 0.167 \cdot \exp(0.223 \cdot \text{thal}) \\
& + 1.723 \cdot \exp(1.15 \cdot \text{thal}) + 0.24 \cdot \exp(1.187 \cdot \text{thal}) \\
& - 0.432 \cdot \Big( 0.002 \cdot \text{age} - 0.005 \cdot \text{ca} + 0.013 \cdot \text{thalach} \\
& \quad + 0.086 \cdot (0.348 - \text{slope})^2 + \exp(1.15 \cdot \text{thal}) - 0.062 \\
& \quad\quad + 0.002/(0.145 - 0.399 \cdot \text{cp}) \Big)^2 \\
& + 4.514 \cdot \sin(6.444 \cdot \text{ca} - 7.642) + 6.5 \cdot \sin(2.913 \cdot \text{cp} + 0.346) \\
& + 0.003 \cdot \sin(9.505 \cdot \text{cp} - 0.282) - 0.289 \cdot \sin(9.937 \cdot \text{cp} - 8.617) \\
& - 0.14 \cdot \sin \Big( 2.564 \cdot \text{age} + 1.14 \cdot \text{oldpeak} + 1.84 \cdot \text{trestbps} \\
& \quad + 0.78 \cdot \text{chol} - 0.489 \cdot \text{thalach} + 7.743 \cdot (-0.309 \cdot \text{ca} - 1)^2 \\
& + 2.452 \cdot \sin(9.505 \cdot \text{cp} - 0.282) + 1.867 \cdot \exp(-6.649 \cdot \text{exang}) + 5.173 \Big) \\
& \quad + 0.003 \cdot \exp(-6.649 \cdot \text{exang}) - 0.343/(-2.648 \cdot \text{ca} - 0.966) \\
& - 0.013/\Big( -0.002 \cdot (-0.744 \cdot \text{age} - 1)^2 - 0.089 \cdot \exp(1.389 \cdot \text{thal}) - 0.011 \Big) \\
& - 0.002/\Big( -0.008 \cdot (-0.744 \cdot \text{age} - 1)^2 - 0.273 \cdot \exp(1.389 \cdot \text{thal}) - 0.001 \Big) \\
& \quad\quad + 0.004/(0.145 - 0.399 \cdot \text{cp}) - 3.978 \quad (16)
\end{aligned}
$$

**Analysis: Why This is a "Spline Soup".** This formula exhibits several pathological characteristics that render it practically uninterpretable:

- **Indiscriminate Feature Mixing:** All 13 features appear scattered throughout the expression with no semantic organization. For example, `age` appears in 5 different contexts (linear term, squared terms, nested sine function), making it impossible to understand its overall contribution.

- **Deep Nesting:** The formula contains 3-4 levels of nested functions (e.g., sine of a sum containing exponentials and squared terms), obscuring causal relationships.

- **Redundant Representations:** The same feature appears in multiple similar forms (e.g., three different exponentials of `thal` with coefficients 0.223, 1.15, 1.187), suggesting the model learned redundant pathways rather than discovering true structure.

- **Arbitrary Function Choices:** The formula mixes 11 different function types (linear, squared, exponential, sine, rational) without clear semantic justification, appearing more like numerical overfitting than knowledge discovery.

**Visualization.** Figure 6 shows the network structure of the trained standard KAN. The dense all-to-all connectivity creates a tangled web where every input feature connects to every hidden node, making it impossible to trace which features interact or identify semantic groupings. This visualization starkly contrasts with HKAN's clean hierarchical structure shown in Figure 3.

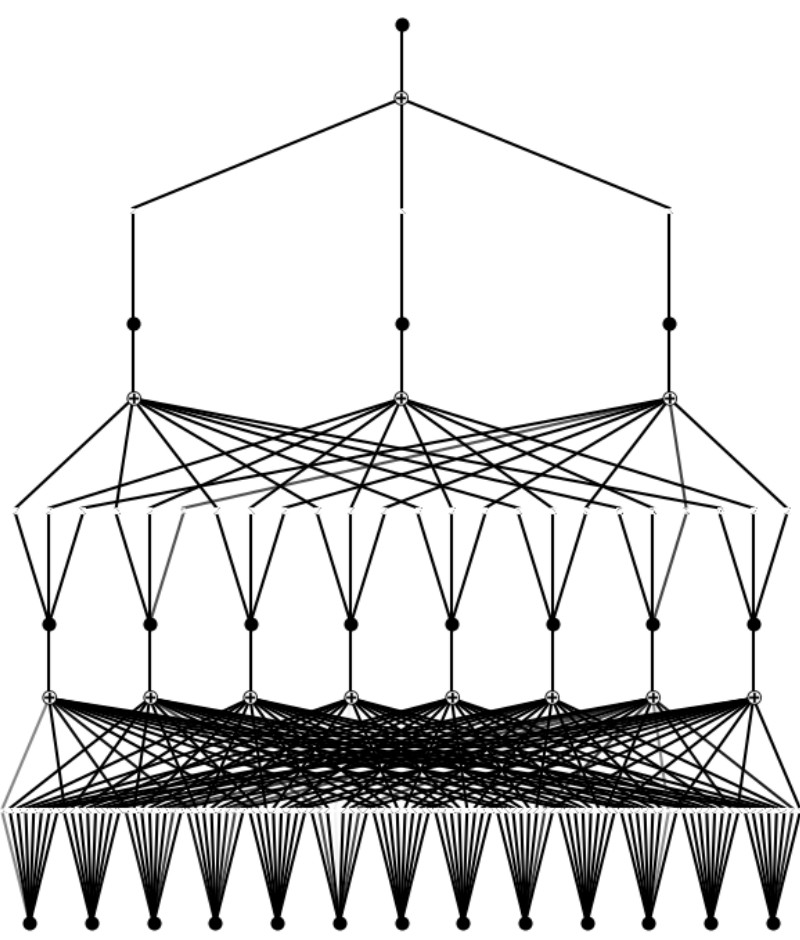

Figure 6: Visualization of standard KAN's dense connectivity on UCI Heart Disease. The all-to-all connections create a "spline soup" where semantic structure is lost in the tangle of interactions. Compare this to HKAN's sparse hierarchical structure in Figure 3.

**Conclusion.** This comparison demonstrates that while standard KAN achieves reasonable predictive performance (0.919 AUC), its dense parameterization produces formulas that are effectively black boxes. HKAN's hierarchical sparse design with evolutionary topology discovery and dual-layer regularization is not merely an optimization—it is a fundamental architectural innovation that transforms KAN from a universal approximator into a practical knowledge discovery tool.

### A.10 FUNCTION FITTING DETAILS

**Case 1: 3D Polynomial Function.** For the simple polynomial $F(x_1, x_2, x_3) = x_1^2 + 5x_2 + x_3^2$, both methods achieve excellent performance:

**Standard KAN:** Achieves perfect symbolic recovery through progressive training with gradually decreasing regularization:

$$F_{\text{KAN}} = 1.0x_1^2 + 5.0x_2 + 1.0x_3^2 \tag{17}$$

**HKAN:** Learns near-perfect coefficients with hierarchical decomposition:

$$F_{\text{HKAN}} = 1.01x_1^2 + 5.01x_2 + 1.00x_3^2 \tag{18}$$

**Case 2: 4D Composite Function.** For the complex function $F = \exp(x_1^2 + x_2^2) + \frac{1}{1+x_3+x_4}$:

**HKAN Factor Decomposition:**

$$f_1(x_1, x_2) = 0.818 \cdot \exp(0.896x_1^2 + 0.898x_2^2) + 1.112 \tag{19}$$

$$f_2(x_3, x_4) = -1.291 - \frac{0.314}{-0.732x_3 - 0.733x_4 - 0.744} \tag{20}$$

$$F_{\text{HKAN}} = 1.395 \cdot f_1 + 2.442 \cdot f_2 + 1.437 \tag{21}$$

**Standard KAN Expression:**

$$F_{\text{KAN}} = -0.768x_3 - 0.773x_4 + 1.221(x_1 + 0.001)^2 + 1.231(-x_2 - 0.003)^2$$
$$+ 7.863\Big(-0.021(0.002 - x_2)^2 - 0.015(x_1 - 0.002)^2 - 1$$
$$- \frac{0.226}{-0.3x_4 - 0.429} - \frac{0.223}{-0.29x_3 - 0.424}\Big)^2 + 1.956 \tag{22}$$

The standard KAN expression spans 202 characters with complex nested terms, while HKAN produces a clean 50-character formula that correctly identifies the exponential and rational components.

### A.11 DATASET DETAILS

Table 15 provides comprehensive statistics for all datasets used in our experiments.

Table 15: Detailed dataset characteristics

| Dataset | Samples | Features | Task | Classes | Domain |
|---|---|---|---|---|---|
| *Small-Scale Datasets* | | | | | |
| UCI Heart Disease | 303 | 13 | Binary Clf. | 2 | Medical |
| Glass | 214 | 9 | Multi-Clf. | 7 | Materials |
| UCI Student | 649 | 32 | Regression | - | Education |
| *Medium-Scale Datasets* | | | | | |
| California Housing | 20,640 | 8 | Regression | - | Real Estate |
| Adult | 48,842 | 14 | Binary Clf. | 2 | Census |
| German Credit | 1,000 | 20 | Binary Clf. | 2 | Finance |
| Higgs | 98,050 | 28 | Binary Clf. | 2 | Physics |
| *Large-Scale Datasets* | | | | | |
| Covtype | 581,012 | 54 | Multi-Clf. | 7 | Ecology |
| HomeCredit Default | 458,913 | 696 | Binary Clf. | 2 | Finance |
| Delivery ETA | 539,577 | 223 | Regression | - | Logistics |

**Evaluation Metrics.**

- Binary classification: AUC-ROC (Area Under the Receiver Operating Characteristic Curve)
- Multi-class classification: Accuracy (percentage of correct predictions)
- Regression: RMSE (Root Mean Squared Error)
- Parameter efficiency: Total number of trainable parameters

**Feature Preprocessing.** All datasets underwent standardization (zero mean, unit variance) for continuous features and label encoding (ordinal encoding) for categorical features. We chose label encoding over one-hot encoding because: (1) HKAN shows minimal performance difference between these encoding schemes due to its adaptive B-spline functions that can learn arbitrary mappings from ordinal values, and (2) using one-hot encoding would unfairly disadvantage other methods in parameter count comparisons. Missing values were imputed using median for numerical features and mode for categorical features. No feature engineering was performed to ensure fair comparison with baselines.

## A.12    LIMITATIONS AND FUTURE DIRECTIONS

While HKAN demonstrates strong performance across diverse scenarios, several limitations warrant acknowledgment. The evolutionary architecture search introduces computational overhead during discovery, and traditional gradient boosting methods like XGBoost maintain advantages on certain large-scale datasets due to extensive optimization for tabular data. Our current evaluation focuses on general tabular data; domain-specific applications may require specialized adaptations.

Future research directions include: (1) applications in recommendation systems where natural feature groupings could provide valuable business insights; (2) extension to computer vision and speech processing by replacing MLP components with interpretable KAN units; (3) algorithmic improvements through gradient-based NAS methods to reduce computational cost; and (4) developing incremental architecture update mechanisms. HKAN represents a significant step toward building transparent and accountable AI systems, bridging the gap between interpretability demands and performance requirements in modern machine learning.

## A.13    LARGE LANGUAGE MODEL USAGE

In accordance with ICLR 2026 guidelines, we disclose the use of Large Language Models in this research. LLMs were employed as general-purpose assistance tools and did not contribute to the core research ideation or methodology development.

**Paper Writing Assistance:** Claude Sonnet 4.0 was used for manuscript refinement, including language polishing, result summarization, and analysis presentation. The core scientific contributions, experimental design, and conclusions remain entirely the work of the authors.

**Code Implementation Support:** Claude Sonnet 4.0 assisted in adapting existing author-developed code to different datasets, primarily for data preprocessing and experimental pipeline setup. All algorithmic innovations and core implementations were developed independently by the authors.

**Visualization Design:** Gemini 2.5 Pro provided RGB color value recommendations for figure design to enhance visual clarity and accessibility.

The authors take full responsibility for all content, including any LLM-generated text that has been reviewed, validated, and integrated into the manuscript.

