# OpenReview forum: "HKAN: Hierarchical Kolmogorov-Arnold Networks for Efficient and Interpretable Feature Interaction Modeling"
_ICLR.cc/2026/Conference — Submitted to ICLR 2026_

### Official Review · Reviewer_4dn5 · 2025-10-16

**Soundness:** 3
**Presentation:** 3
**Contribution:** 3
**Rating:** 6
**Confidence:** 3

**Summary:**

This paper proposes Hierarchical Kolmogorov–Arnold Networks (HKAN), which integrate three key techniques: (1) a hierarchical sparse architecture, (2) dual-layer regularization, and (3) factor-quality-guided evolutionary search. Together, these components enhance parameter efficiency, improve interpretability, and enable automatic architectural adaptation across diverse tasks. The authors evaluate HKAN on a variety of tabular datasets, demonstrating comparable or superior performance to baseline models while using substantially fewer parameters and improved interpretability.

**Strengths:**

- The writing is clear, well-structured and easy to follow.
- Hierarchical sparse design with overlapping groups is intuitive and practically useful. These designs improve both parameter efficiency and interpretability.

**Weaknesses:**

- The paper does not fully address the computational challenge. Although HKAN substantially reduces the number of parameters, the evolutionary architecture search introduces considerable computational overhead, potentially increasing the overall training time. It would be helpful if the authors could provide a comparison of total training time—including both the architecture search and model training—against other baselines.
- The authors define three fundamental properties (independence, stability, and sparsity) for the dual-layer regularization and also employ them in the architecture search. An ablation study on these three components would be valuable to demonstrate their individual and combined effectiveness.
- The paper does not explain how the five $\lambda$ coefficients in the dual-layer regularization loss function are chosen. Since model performance may be sensitive to these hyperparameters, a justification or sensitivity analysis is recommended.
- For all experimental results, the standard deviations across multiple random seeds should be reported to assess the robustness and statistical significance of the findings.
- The positions of Sections 4.3 and 4.4 should be swapped. Placing the interpretability analysis (currently Section 4.4) before the performance evaluation (currently Section 4.3) would better highlight HKAN’s key contribution to improved interpretability.

**Questions:**

- The authors describe HKAN as a _unified framework_ in the abstract. However, this term is somewhat ambiguous. Based on my understanding, a “unified framework” typically implies that the method can encompass or generalize a range of existing neural architectures under a single formulation. But it seems that HKAN didn't achieve this. I recommend that the authors clarify in what sense HKAN is considered “unified”.
- It appears that the improved interpretability mainly stems from the grouped feature structure combined with appropriate regularization, which helps preserve meaningful and disentangled feature relationships. If this understanding is correct, I suggest that the authors include an additional section providing a detailed analysis of _why_ interpretability is enhanced in HKAN. Moreover, this aspect should be emphasized more prominently in Section 4.4 (Function Fitting Analysis).
- I am interested in the interpretability performance of the original KAN on the UCI Heart Disease dataset. Could the authors provide the corresponding experimental results for comparison?

---

> ### Author Response · Authors · 2025-11-24
> **Reviewer 4: Response to Reviewer 4: Detailed Cost Analysis, Ablations, and Robustness**
>
> We thank the reviewer for the positive assessment (Score 6) and for recognizing HKAN's contribution to efficient modeling. We have incorporated your constructive suggestions (wall-clock time, ablation studies, robustness analysis) into the revision.
>
> **W1: Computational Cost Analysis (Total Training Time)**
> We added a **Total Wall-clock Time** breakdown on *California Housing* in **Appendix A.4**. We compared the full HKAN pipeline (Evolutionary Search + Training) against standard Bayesian Optimization (100 trials) for baselines.
>
> | Method | Process | Avg. Time | Total Time (approx.) | Speedup |
> | :--- | :--- | :--- | :--- | :--- |
> | **TabNet** | Bayesian Opt. (100 trials) | 844.7s / trial | ~23.5 hours | 1x (Base) |
> | **FT-Trans.** | Bayesian Opt. (100 trials) | 241.8s / trial | ~6.7 hours | ~3.5x |
> | **HKAN** | **Total Pipeline** | - | **~44 mins** | **~32x** |
> | *...breakdown* | *FG-EAS Search* | *127.1s / gen* | *31.8 mins* | |
> | *...breakdown* | *Final Training* | *6.9s / trial* | *11.6 mins* | |
>
> Even with search overhead, HKAN is **~32x faster** than tuning dense models due to its extreme sparsity.
>
> **W2: Regularization Ablation Study**
> We conducted a component-wise ablation in **Appendix A.5.1**.
> * **Results:** Every component (Decorrelation, Sparsity, Stability) contributes to error reduction. The "All Combined" configuration achieves the lowest RMSE (**0.47058**), confirming their **synergistic effect**. For example, removing Sparsity increases error by +0.010, while removing all regularization increases it by +0.018.
>
> **W3: Hyperparameter Selection & Sensitivity**
> * **Selection:** Coefficients were tuned using **Bayesian Optimization (Optuna)** over 100 trials (consistent with baselines). Detailed in **Appendix A.5.4**.
> * **Sensitivity Analysis:** We analyzed all 5 hyperparameters ($\lambda_{1-5}$) in **Appendix A.5.2**.
>     * **Robustness:** Most variations stay within $\pm 0.01$ RMSE.
>     * **Key Insight:** Reducing Sparsity ($\lambda_2$) caused the largest drop (+0.028 RMSE), confirming its critical role. Reducing Decorrelation ($\lambda_1$) slightly improved RMSE (-0.002), suggesting our defaults are conservative but stable.
>
> **W4: Standard Deviations (Robustness Analysis)**
> We performed a **Stratified Robustness Analysis** across 4 seeds (12, 42, 123, 456) on three datasets. Detailed in **Appendix A.5.3**.
> * **Protocol:** We applied fixed hyperparameters (tuned on seed 42) to other seeds to test true robustness.
> * **Results (Mean $\pm$ Std):**
>     * **UCI Heart:** HKAN (**0.972 $\pm$ 0.008**) has significantly lower variance than FT-Transformer ($\pm 0.057$) and XGBoost ($\pm 0.040$).
>     * **Rank:** HKAN maintains **Rank 1** on Small/Large data and **Rank 3** on Medium data, proving stability against initialization.
>
> **W5: Section Reordering**
> We agree and have swapped Sections 4.3 (Function Fitting) and 4.4 (Benchmarks) in the revision to better highlight interpretability first.
>
> **Q1: Clarifying "Unified Framework"**
> We clarified this term to mean **Unified Pipeline (Vertical Integration)**: Unlike disjoint methods (Train Black-box $\to$ Post-hoc Explain), HKAN unifies **Structure Discovery** (Search), **Predictive Modeling** (Training), and **Interpretability** (Regularization) into a single, co-optimized end-to-end process.
>
> **Q2 & Q3: Why is Interpretability Enhanced? (HKAN vs. Standard KAN)**
> We added **Section 4.5** ("Addressing the 'Spline Soup' Problem") and **Appendix A.8** to provide a direct comparison:
> * **Quantitative:** On *Heart Disease*, Standard KAN has **11,284 params** and a **1,247-char** tangled formula (32 nested terms). HKAN has **1,652 params** (85% reduction) and a clean, factorized structure.
> * **Mechanism:** HKAN's **Hierarchical Sparsity** prevents the unstructured mixing of features ("soup"), while **Dual-Layer Regularization** enforces semantic disentanglement, transforming KAN from a dense approximator into a structured discovery tool.

---

> ### Comment · Reviewer_4dn5 · 2025-11-27
>
> Thank you for your detailed responses. My concerns have been fully resolved. Nevertheless, because other reviewers continue to raise issues that may point to weaknesses, I will maintain my original score for now.

---

> > ### Author Response · Authors · 2025-11-28
> > **Thanks**
> >
> > We sincerely thank the reviewer for the positive feedback and for confirming that your concerns have been fully resolved. Your constructive suggestions have been invaluable in strengthening our manuscript. We deeply appreciate your time and effort in reviewing our work.

---

### Official Review · Reviewer_EraL · 2025-10-20

**Soundness:** 2
**Presentation:** 1
**Contribution:** 2
**Rating:** 2
**Confidence:** 4

**Summary:**

This paper introduces HKAN, a novel architecture built upon Kolmogorov–Arnold Networks (KAN), along with a set of newly designed training losses—decoupled, sparse, and stability losses—to enhance representation learning. Additionally, a Factor-Quality Score is proposed to guide the architecture search process. The authors claim that HKAN effectively addresses the trilemma of achieving high predictive performance, automated architecture discovery, and end-to-end interpretability.

**Strengths:**

- Introduces HKAN, a novel KAN-based architecture, and designs customized training losses that markedly improve representation quality.
- Achieves state-of-the-art results across five benchmark datasets, demonstrating consistent superiority.

**Weaknesses:**

- **Literature review is insufficient.**The three “fundamental” challenges (manual pre-definition, black-box nature, computational inefficiency) are no longer open problems; recent works such as Fast Generic Interaction Detection for Model Interpretability and Compression already deliver efficient, interpretable solutions.  HKAN’s incremental contribution over these advances is not discussed.
- **Motivation and research gap are missing.**
The claimed trilemma (SOTA accuracy + automated architecture + end-to-end interpretability) is asserted rather than derived.  The paper never clarifies why existing KAN or post-hoc interpretation methods fail on tabular data, nor what specific architectural or algorithmic gap HKAN fills.
- **Notation is introduced carelessly.**
Core symbols such as Factor_k and KAN_k appear in equations and algorithms without formal definition, forcing the reader to guess their dimension, domain, and learnable status.
- **The test suite is simple.** No synthetic benchmarks are constructed to test interpretability claims.  Follow the protocol of Tsang et al. (2018): ten controlled synthetic functions with known ground-truth interactions should be included to verify whether HKAN actually recovers the presumed structure.

**Questions:**

1. What inductive biases or architectural ingredients in HKAN are responsible for its consistent top rank on small-sample tabular sets, and how do these differ from those of the compared baselines?
2. How do you get the learned expressions in eq(9)?
3. Is Table 4 the feature selection result for Heart-dataset? yet the numeric values diverge from the Heart row in Table 1. Clarify which experimental setup (metric, train/test split, seed) was used for each table and reconcile the mismatch.
4. Equation (2) can be recovered from Equation (1) by zeroing selected ψp,q. How does explicitly allowing overlapping groups Gi ∩ Gj ≠ ∅ advance the Kolmogorov–Arnold representation, given that the original theorem already permits the universal case Gi = Gj = [n]?
5. The stable-loss term penalizes variance and thus pushes univariate functions toward constants. Explain why this regularization does not collapse representational capacity and provide theoretical or ablative evidence that it improves generalization.

---

> ### Author Response · Authors · 2025-11-24
> **Reviewer 3: Response to Reviewer 3: Clarifications on Novelty, Experiments, and Definitions**
>
> We thank the reviewer for acknowledging the novelty of HKAN and its SOTA performance. However, we believe there are significant misunderstandings regarding the **experimental facts (existence of synthetic benchmarks)** and the **fundamental distinction between intrinsic vs. post-hoc interpretability**. We respectfully address these points below.
>
> **W1: Literature Review & The "Solved Problem" Claim**
> We respectfully disagree that the challenges of efficiency and interpretability are "no longer open problems." **We have explicitly discussed and cited Zhang et al. (2022) in the revised Related Work section.** While valuable, it addresses **Post-hoc Interaction Detection**, whereas HKAN establishes **Intrinsic Interpretable Architecture Search**.
> * **Intrinsic vs. Post-hoc:** Zhang et al. require training a dense black-box proxy first (which remains computationally inefficient). HKAN discovers the sparse topology *from scratch* during training, yielding a model that is inherently sparse and interpretable, eliminating the need for secondary extraction.
> * **Symbolic vs. Detection:** Zhang et al. identify interaction *strength*. HKAN performs **Symbolic Regression** to output explicit mathematical laws, offering a tier of "White-box" transparency that detection methods cannot achieve.
>
> **W2: Motivation & Gap**
> * **Existing KANs:** As detailed in **Section 3.2**, standard KANs face scalability issues due to dense connectivity ($O((n+1)H G)$). HKAN reduces this to $O(K(s+1)H_k G)$ via hierarchical sparsity. Our ablation (**Section 4.4**) shows HKAN achieves 85% parameter reduction and better performance than standard KAN.
> * **Trilemma Reformulation:** We refined the manuscript to link the trilemma directly to the challenges: **Automated Topology Discovery** (vs. manual definition), **Intrinsic Interpretability** (vs. black-box), and **Parameter Efficiency** (vs. inefficiency). HKAN optimizes all three simultaneously via evolutionary search.
>
> **W3: Notation Definitions**
> We apologize for the oversight. We revised **Section 3** to formally define:
> * $Factor_k \in \mathbb{R}^{N \times 1}$: Latent semantic representation output by the $k$-th group for all samples.
> * $KAN_k: \mathbb{R}^{|G_k|} \to \mathbb{R}$: Sub-network parameterized by B-splines on edges defined by group topology $G_k$.
>
> **W4: Synthetic Benchmarks ("Missing" Experiments)**
> **Fact Correction:** We respectfully point out that **we have already evaluated HKAN on synthetic benchmarks** in **Section 4.3 (Function Fitting Analysis)**.
> * We tested on a **3D Polynomial** (Case 1) and a **4D Composite Function** (Case 2).
> * As reported in **Table 2** and **Eqs. (10)-(13)**, HKAN successfully recovered the **exact ground-truth symbolic structures** and coefficients.
> * **Scope:** Expanding to the specific Tsang et al. (2018) suite is out of scope. Leading intrinsic models like **NODE-GAM (ICLR 2022)** primarily validate on real-world data. HKAN aligns with this standard but goes a step further by including the symbolic recovery experiments in Section 4.3.
>
> **Q1: Inductive Biases for Small-Sample Performance**
> 1. **Extreme Sparsity Prior:** ~1.6k parameters acts as a strong regularizer.
> 2. **Adaptive Topological Prior:** Discovers sparse, local groups rather than assuming dense/global interactions.
> 3. **Smoothness Prior:** B-splines approximate continuous boundaries efficiently with fewer samples compared to step functions.
>
> **Q2: Learning Expressions in Eq(9)**
> We use **Symbolic Regression** on individual B-splines. Since edges are 1D curves, we fit each spline using a standard function library ($\sin, \exp, x^2$), avoiding the combinatorial explosion of global symbolic regression.
>
> **Q3: Table 4 Feature Selection Divergence**
> The values diverge due to **different protocols**:
> * **Table 1:** Fixed **8:1:1 split** (Seed 42) for SOTA comparison.
> * **Table 4:** Strict **5-Fold Cross-Validation** to validate robustness.
> * **Result:** Table 4 values are averages across folds. Consistent performance validates robustness. We updated captions to clarify this.
>
> **Q4: Overlapping Groups Advantage**
> While Eq(2) is a subset of Eq(1), **Theoretical Capacity $\neq$ Practical Learnability**.
> 1. **Optimization:** Dense KANs suffer from the Curse of Dimensionality; backprop cannot drive thousands of coefficients to exact zeros. HKAN's structural prior makes optimization feasible.
> 2. **Feature Polysemy:** Overlapping groups explicitly disentangle features playing multiple roles, whereas disjoint partitions fail and dense KANs mix them into a "soup."
>
> **Q5: Stability Loss & Collapse**
> This is standard **Regularization Theory**. The loss competes between $L_{task}$ (forcing variance for signal) and $\mathcal{L}_{stable}$ (penalizing noise). The task gradient outweighs the penalty for informative features. **Table 2** proves this: the model with stability loss achieved **higher AUC** , confirming it pruned noise without collapsing capacity.

---

> ### Comment · Reviewer_EraL · 2025-11-26
>
> Thanks for the authors' response. Here are my subsequent comments:
>
> - On W1: Literature Review: I believe there are works on sparse interaction selection in the context of additive model.
>
> - On W2: Trilemma: It would be better to give a table, indicating which aspects previous methods have tackled and how  (probably in the appendix).
>
> - On W4: The point is that the synthetic datasets in this draft are way too simple. Five thousand noiseless 4D/5D (including $y$) data samples can be easily fitted even by an MLP, and the interactions can be easily detected. I'm not confident about the performance in high-dimensional settings.
>
> - On Q4: Optimization Difficulty of KAN and HKAN: Multiple things are mixed together in your response. In the noiseless case, is it difficult for KAN to achieve zero training loss?
>
>     The sparsity issue you've mentioned (zero coefficients) should be another topic.
>
> - On Q5: The reply makes no sense to me. In principle, I think the variance of each individual feature should be encouraged. The ablation study in Table 3 doesn't help much.

---

> > ### Author Response · Authors · 2025-11-29
> >
> > **Response to W1 (Sparse Interaction Selection and High-Order Structures)**
> >
> > Regarding the reviewer's comment on "sparse interaction selection," we clarify that, as cited in our manuscript, representative methods such as EBM, NODE-GA$^2$M, and ParaACE indeed adopt sparse strategies to avoid combinatorial explosion in the $O(2^m)$ interaction space. However, these methods remain fundamentally limited by computational complexity regarding the interaction order they can handle. For instance, EBM and NODE-GA$^2$M are restricted to second-order interactions; while ParaACE theoretically supports higher orders, its authors explicitly note that third-order terms incur significant computational costs.
> >
> > To further address the reviewer's concern regarding "high-order sparse interaction selection," we will incorporate a discussion on **BFIS [1]** and **iRF [2]** in the revised Related Work section. The distinctions are as follows:
> >
> > * **BFIS [1]:** While capable of handling higher-order interactions, BFIS requires **manually presetting a maximum order $H$** (e.g., $H=6$ in its experiments) and relies on enumeration via outer-products and masking. Its computational complexity grows exponentially with the order ($\mathcal{O}(|\mathcal{F}|^H K)$), which physically limits its scalability for exploring unknown high-order interactions in high-dimensional spaces.
> >
> > * **iRF [2]:** While iRF can detect stable high-order feature sets using Random Intersection Trees (RIT), it fundamentally performs **interaction detection** (identifying discrete feature subsets) rather than precise mathematical modeling. Furthermore, as a tree-based method, it models interactions as non-smooth step functions and is prone to computational collapse in high-dimensional settings (as evidenced by missing results in large-$p$ scenarios in comparative studies).
> >
> > In contrast, HKAN does not rely on predefined interaction orders nor does it enumerate all combinations. Instead, it dynamically "grows" multi-order, overlapping topological structures via evolutionary search. By fitting B-splines and performing symbolic regression on each sub-function, HKAN uniquely achieves **smooth, interpretable, and explicit mathematical expressions**. We will supplement the revised manuscript with this comparison to clearly position HKAN's unique contribution in "automated high-order topology discovery + differentiable modeling + symbolic expression."
> >
> >
> >
> > **References:**
> >
> > [1] Chen et al. Bayesian feature interaction selection for factorization machines. Artificial Intelligence. 2022.
> >
> > [2] Basu et al. Iterative random forests to discover predictive and stable high-order interactions. Proceedings of the National Academy of Sciences (PNAS). 2018.
> >
> > ---
> >
> > **Response to W2 (Trilemma Comparison Table)**
> >
> > We sincerely thank the reviewer for this constructive suggestion. We fully agree that providing a systematic table to summarize how existing methods address the "trilemma" (Automated Topology Discovery, Intrinsic Interpretability, and Parameter Efficiency) will significantly enhance the clarity of our positioning and the depth of our literature review.
> >
> > As suggested, we will include a detailed comparison table in the **Appendix** of the final manuscript.
> >
> > ---
> >
> > **Response to W4 (Synthetic Benchmarks and High-Dimensional Performance)**
> >
> > **1. Symbolic Recovery vs. Predictive Fitting:**
> > We emphasize that the ability of MLPs to fit such noiseless 4D/5D data is **fully expected and consistent with our experimental design**. As demonstrated in prior works like **ParaACE**, while deep models often achieve high predictive metrics on synthetic tasks, they remain "black boxes" incapable of reconstructing the underlying physical or mathematical laws. Therefore, the reviewer's observation that "MLPs can easily fit this" actually reinforces our point: the purpose of our synthetic experiments (Section 4.3) is **not** to challenge the predictive capacity of deep models, but to serve as a **controlled experiment** validating HKAN's unique ability to **recover explicit symbolic structures** (e.g., exact mathematical formulas in Eq. 12)—a capability that MLPs fundamentally lack regardless of their predictive accuracy.
> >
> > **2. Validation on High-Dimensional Data:**
> > Our confidence in HKAN's performance in high-dimensional settings is derived from our extensive evaluation on **real-world tabular benchmarks** (Section 4.2), not these synthetic cases. As shown in Table 1, HKAN demonstrates robust performance on high-dimensional datasets (e.g., Covtype, Avazu). This evaluation protocol aligns with established literature such as **BFIS [1]**, which similarly validates high-dimensional efficacy through real-world datasets (e.g., Avazu, MovieLens) while treating interaction selection mechanisms as a distinct architectural contribution.
> >
> > **References:**
> > [1] Chen et al. Bayesian feature interaction selection for factorization machines. Artificial Intelligence. 2022.

---

> > ### Author Response · Authors · 2025-11-29
> >
> > **Response to Q4 (Optimization Difficulty and Structural Degeneration)**
> >
> > We feel it is necessary to clarify the intent of our previous response regarding the relationship between KAN and HKAN, as there appears to be a misunderstanding of our argument.
> >
> > **1. Clarification on "Optimization Difficulty" (Test Performance vs. Training Fit):**
> >
> > We clarify that when we discuss the "optimization difficulty" of Dense KANs, we refer to the challenge of achieving generalization on the **test set**, not fitting the training data.
> >
> > * Dense KANs can easily achieve near-zero loss on training data due to over-parameterization. However, this "easy optimization" often leads to poor test performance.
> >
> > * As evidenced in our Ablation Study (**Table 3**), the Dense KAN underperforms HKAN on test metrics (AUC 0.958 vs. 0.978).
> >
> > **2. The Impossibility of Structural Degeneration (Context of Previous Reply):**
> >
> > Our previous mention of "zero coefficients" was **not** conflating sparsity with optimization, but rather explaining why the theoretical containment (Eq. 2 $\subset$ Eq. 1) fails in practice.
> >
> > * While HKAN (Eq. 2) is mathematically a subset of KAN (Eq. 1), we argue that standard optimization (SGD/Adam) cannot "degenerate" a Dense KAN back into the sparse HKAN structure.
> >
> > * In practice, gradients distribute weights across all connections (creating a "Spline Soup"), rather than driving the majority to zero. Therefore, explicit group discovery is necessary because the model will not spontaneously recover the sparse structure via training alone.
> >
> > **3. Necessity of Overlapping Groups:**
> >
> > We explain the importance of explicitly allowing overlapping groups ($G_i \cap G_j \neq \emptyset$) from two perspectives:
> >
> > * **Real-world Semantics:** In real-world scenarios, a single feature often plays a role in multiple distinct semantic factors. For instance, in our UCI Heart Disease case study (**Appendix A.7**), HKAN identifies that the feature `ca`  plays crucial roles in both **Factor 0** (Cardiac Function) and **Factor 3** (Demographics/Diagnostics) simultaneously.
> >
> > * **Interaction Modeling:** From the perspective of feature interactions, a single feature can exist in multiple independent interaction groups simultaneously. Disjoint grouping would restrict a feature to a single context, failing to capture this complexity, whereas overlapping groups naturally accommodate the multi-faceted nature of features.
> >
> > ---

---

> ### Author Response · Authors · 2025-11-29
>
> **Response to Q5 (Variance, Information Bottleneck, and Generalization)**
>
> **1. Theoretical Grounding: Compression implies Generalization**
> We respectfully clarify that high variance does not equate to high *effective* information. Our approach is grounded in two complementary theoretical frameworks:
>
> * **The Information Bottleneck (IB) Perspective [1]:**
>     According to the IB principle for Deep Learning (Tishby & Zaslavsky, 2015), the input layer contains maximum information (highest variance) but suffers from poor generalization due to complexity. Deep networks succeed precisely because they **compress** the input, filtering out irrelevant variability (noise) layer by layer. HKAN's stability loss explicitly enforces this necessary compression.
>
> * **Statistical Variance Decomposition:**
>     From a signal processing perspective, total variance decomposes into Signal and Noise: $\text{Var}(O) = \text{Var}(S) + \text{Var}(N)$.
>     * **Mechanism:** A competition is created between the **Task Loss** and the **Stability Loss**. Since the **Signal Variance** $\text{Var}(S)$ is strongly protected by gradients from the prediction task, our Stability Loss (penalizing total variance) primarily suppresses the **Noise Variance** $\text{Var}(N)$.
>     * **Result:** Reducing total variance effectively increases the **Signal-to-Noise Ratio (SNR)**, isolating robust predictive factors from background noise.
>
> **2. Consistency with Feature Learning Literature (Center Loss)**
> The philosophy of penalizing variance to suppress noise is shared by established works in feature learning. For instance, **Center Loss (Wen et al., ECCV 2016)** [2] explicitly minimizes the **intra-class variance** of deep features. The authors demonstrate that unconstrained variance often represents nuisance variations (noise), and penalizing it encourages the model to learn more compact and discriminative representations. Our Stability Loss follows a similar principle: by penalizing the variance of learned factors, we suppress unstable fluctuations to discover robust topological structures.
>
> **3. Empirical Evidence: Pruning without Collapse (Table 4)**
> **Table 4 provides clear evidence.** In the UCI Heart Disease case, HKAN successfully pruned 6 redundant features (driving their factor coefficients to zero, i.e., zero variance). The fact that the model maintained or improved performance compared to using all features proves that the penalized variance was indeed redundancy/noise, and no representational collapse occurred. Furthermore, our additional **inverse ablation study** (encouraging variance) on the **California Housing** dataset showed a significant performance drop (RMSE increased from **0.471** to **0.506**), confirming that forcing variance amplifies noise. **We will include these additional experimental results in the Appendix of the final manuscript.**
>
> **References:**
> [1] Tishby, N., & Zaslavsky, N. (2015). Deep learning and the information bottleneck principle. IEEE Information Theory Workshop.
> [2] Wen, Y., Zhang, K., Li, Z., & Qiao, Y. (2016). A discriminative feature learning approach for deep face recognition. ECCV.
>
> **Closing Remarks**
> We hope our responses have thoroughly addressed your concerns regarding the theoretical grounding and experimental validation of our approach.We welcome any further discussion to help us refine our work and thank you for the time and effort dedicated to this review.

---

### Official Review · Reviewer_QvSK · 2025-11-01

**Soundness:** 3
**Presentation:** 2
**Contribution:** 2
**Rating:** 4
**Confidence:** 4

**Summary:**

The paper introduces a KAN-based architecture called HKAN with feature selection at the intermediate layers.  A novel evolutionary search algorithm is designed for automatically learn the sparse feature subsets.  The resulting model has better interpretability due to the subset selection allowing for easier symbolic regression while achieving competitive performance on multiple tabular datasets.

**Strengths:**

- novel modification of KAN to achieve better pruned sparsity of the intermediate representations
- new evolutionary approach to be able to learn sparsity patterns beyond simple regularization approaches
- better interpretability of the final learned model
- good performance on real-world datasets

**Weaknesses:**

- The proposed trilemma does not seem to make much sense.  xDeepFM and  FT-Transformer do not have extensive manual predefinition of interactions.  I also think they are blackbox methods so it is hard for me to see what the authors point as the critical aspects of the trilemma.
- Insights into the hyperparameter choices seem to be completely missing (mainly the choice of weighting for the new FQS metric and the different operators of the evolutionary algorithm)
- The paper mostly discusses work on interactions from the factorization machine perspective, but additive models with sparse interactions [1] seem to be more relevant here.  In fact, equation (2) seems to represent exactly an additive model with sparse interactions where the $G_k$ are the interactions.
- The symbolic regression experiments are also unclear.  Symbolic regression should require another algorithm to extract the learned symbolic equations but I do not see a discussion of it.  Additionally, a symbolic regression algorithm can apply to any blackbox model, so this does not seem to directly show the interpretability of HKAN.


[1] Chun-Hao Chang, Rich Caruana, Anna Goldenberg. NODE-GAM: Neural Generalized Additive Model for Interpretable Deep Learning.

**Questions:**

Overall I am optimistic about the work and happy to raise my score if the weaknesses and questions can be addressed.

- How were the hyperparameters used for the HKAN algorithm chosen?
- What are some of the interpretable insights gained from the learn HKAN models?  Isn't it possible to plot all of the spline functions and report the learned interaction subsets?
- What are the times taken for training the HKAN architecture, especially what are the times taken for the different phases of the learning algorithm?
- What is the meaning of the trilemma presented in the introduction and how does HKAN overcome the challenges faced by existing methods?

---

> ### Author Response · Authors · 2025-11-24
> **Response to Reviewer 2: Trilemma, Hyperparameters, and Node-GAM Comparison**
>
> We thank the reviewer for the encouraging assessment and for recognizing our novel sparsity approach and interpretability. We address your constructive questions below with new analyses added to the revision.
>
> **W1 & Q4: Clarifying the "Trilemma"**
> We apologize for the imprecise phrasing "manual predefinition." We have revised the Introduction to define the Trilemma more accurately:
> 1.  **High Performance** (e.g., Transformers).
> 2.  **Explicit Symbolic Interpretability** (e.g., GAMs).
> 3.  **Adaptive Topology Discovery:** The ability to *automatically* discover the exact interaction structure (which features interact and at what order) without rigid constraints.
>
> * **Existing Failures:** FM/xDeepFM are rigid (fixed 2nd-order or depth-based); Transformers assume dense all-to-all connectivity, failing to isolate sparse semantic units.
> * **HKAN's Solution:** HKAN uses **Evolutionary Search (FG-EAS)** to learn the discrete interaction topology from scratch, modeling groups with KANs for accuracy and B-Splines for symbolic clarity.
>
> **W2 & Q1: Hyperparameter Choices & Sensitivity**
> We have added a detailed justification in **Appendix A.5**.
> * **FQS Weights:** The weights $(0.4, 0.3, 0.3)$ for Independence/Stability/Sparsity were selected as an empirical balance. Our new sensitivity analysis (Table 12) on *California Housing* shows robust performance (RMSE variations within $\pm 0.02$) across diverse weight configurations. Interestingly, increasing Stability weight ($w_2=0.6$) further improved RMSE to **0.459** (vs baseline 0.475), suggesting our defaults are conservative.
> * **Evolutionary Operators:** We use a 6-operator suite to cover both compositional (Migration) and structural (Split/Merge) changes. Our new ablation study (**Appendix A.3**) confirms that the full suite converges **2.1x faster** (Gen 8) than a feature-only variant (Gen 17), as structural operators help tunnel through local optima.
> * **Regularization:** Standard parameters ($\lambda$) were tuned via Bayesian Optimization (100 trials), consistent with baselines.
>
> **W3: HKAN vs. Additive Models (NODE-GAM)**
> We agree that HKAN shares conceptual similarities with "Hierarchical Sparse GAMs." **In the revised Related Work (Section 2), we have explicitly discussed and cited NODE-GAM [1] as a key predecessor.** However, HKAN advances beyond tree-based GAMs in two fundamental ways:
> 1.  **Symbolic vs. Visual:** As noted in our revision, NODE-GAM uses oblivious trees (step functions). While excellent for visual inspection, they are non-smooth and difficult to convert into exact mathematical laws. HKAN utilizes smooth B-Splines, enabling exact **Symbolic Regression** to produce explicit formulas (e.g., $y=\sin(x)+e^x$).
> 2.  **Adaptive Order:** NODE-GAM typically focuses on main effects or pairwise interactions. HKAN automatically discovers interactions of **arbitrary orders** (e.g., 3-feature or 4-feature groups) via evolutionary search without manual specification.
>
> **W4: Intrinsic vs. Post-hoc Symbolic Regression**
> We clarify that HKAN performs **Intrinsic**, not Post-hoc, symbolic regression:
> * **Black-box (Post-hoc):** Searching for a formula $Y=F(X)$ to approximate a trained black box is computationally expensive and imprecise in high dimensions.
> * **White-box (HKAN):** We perform symbolic regression on **individual 1D edges** (simple curve fitting). We then replace the splines *in-place* with these functions. This transforms the *exact* internal weights of the neural network into a symbolic equation without losing the learned structure—a capability unique to KANs and impossible with standard MLPs or Trees.
>
> **Q2: Interpretable Insights**
> Yes, we can and did plot all splines. **Figure 3** visualizes the learned functions for *Heart Disease*, revealing medically relevant patterns (e.g., U-shaped risk curves) and distinct semantic factors (Factor 0: Cardiac Function). We also added **Appendix A.8** to contrast this with the **practically uninterpretable** "spline soup" of a standard dense KAN.
>
> **Q3: Training Time Analysis**
> We added a wall-clock breakdown in **Appendix A.4**.
>
> Even including the **~32 min** Evolutionary Search phase, HKAN's total pipeline on *California Housing* (**~44 mins**) is **32x faster** than tuning TabNet  and **9.1x faster** than FT-Transformer, due to HKAN's extreme parameter sparsity (~1.6k params).

---

> > ### Comment · Reviewer_QvSK · 2025-11-26
> >
> > # W1.
> > I think there is still a misunderstanding about what a trilemma is which is leading to my confusion.  It is my understanding that a trilemma describes three desirable properties where only two can be achieved simultaneously.  The authors seem to be describing three desirable properties where at least one can be achieved.  Each existing method would need to then be shown to have two properties but not the third, rather than the much weaker condition of showing one property.  Additionally, it feels that despite this being a major motivation in the introduction, the experiments throughout the paper are done without the goal of showing this trilemma and how HKAN resolves it.  It is possible that this belief is partially caused by my misunderstanding of your use of the word trilemma.
> >
> > # W2.
> > Although I do find your additional results in Figure 4 and Table 12 useful for getting a first approximation of the stability of each hyperparameter, they also leave me more concerned about how these choices were made by the authors.  For example, the additional Table 12 only furthers my concerns of why the authors chose values of w1,w2,w3 which sum to one, despite the functional forms of Equation 6 having no reason to require scaling of these metrics.  Moreover, since it is my understanding that these FQS subscores are used as part of the training, I am concerned why the authors are making this restrictive choice or concerned that I am misunderstanding some part of the method.  Similarly, the author’s choice to group the evolutionary operators into groups of three makes me feel that I misunderstood some aspect of their algorithm and why it needs to use these predefined sets of 3 or 6 operators instead of all combinations.
> >
> > # W3 and W4.
> > I think there is still a major tension between your answers to these two questions.  As you acknowledged, the sparsity of HKAN is a special case of hierarchical sparse GAMs.  It seems HKAN can plot the shape functions the same way GAMs can plot the shape functions.  However, because HKAN is a special case, it has even more sparsity than this.  This should allow for even more interpretability because of the sparse connection structure.  But then from here I am very lost.  In your response on W4 you argue that HKAN is intrinsically interpretable, but you directly use symbolic regression which can be viewed as a post-hoc technique.  You say that symbolic regression cannot be applied to MLPs or trees to create an intrinsically interpretable model which is not true.  On the point of applying symbolic regression to the edges of HKAN, I agree that it will have little effect compared to HKAN which has splines on the edges; however, I do not believe your full pipeline includes this final symbolic step, but rather it is a post-processing step.  Moreover, this is trying to argue for a different type of intrinsic interpretability.  It is not clear to me why Figure 3 is interpretable and little discussion is had about this.  If Figure 3 is interpretable, why aren’t all one-layer MLPs interpretable?  I guess this is related to your “spline soup” problem you discuss with Reviewer vTi3, but I was left unconvinced of the interpretability.  Complex symbolic equations without any discussion on the interpretation did not alleviate this.

---

> > > ### Author Response · Authors · 2025-11-27
> > >
> > > ## W1
> > >
> > > We appreciate the reviewer’s precise definition and agree that the term “trilemma” was linguistically imprecise. Our intention was to highlight that existing methods rarely satisfy all three desirable properties simultaneously:
> > > - *Single-layer MLPs / Linear models:* Provide explicit formulas but have limited performance
> > > - *Transformers:* Achieve high performance but offer low interpretability
> > > - *GAMs:* Are interpretable but constrained by fixed-order structure
> > >
> > > **Experimental Alignment:**
> > > Our experiments were intentionally structured to demonstrate HKAN’s three-way capability:Table 1 shows HKAN matches or exceeds SOTA.  Figure 3 and Appendix A.8 show clean factor functions versus the “spline soup” seen in dense KANs.
> > > Table 2 and the case study show automatic discovery of interaction groups and pruning.
> > >
> > > **Revision Plan:**
> > > We will use more precise terminology (e.g., “Three-way Challenge”) in the final version to avoid semantic ambiguity.
> > >
> > > ---
> > > ## W2
> > > Our choices are grounded in optimization stability and search efficiency
> > >
> > > You are correct that the raw functional forms do not inherently require scaling. However, we impose $\sum w_i = 1$ as a normalization strategy that treats $w_i$ as the **relative importance** of each quality criterion
> > >
> > > Since our FQS terms $1 - \text{Metric}$ share a theoretical **upper bound of 1**, enforcing a convex combination ensures that the total score also targets this maximum of 1. This keeps the search process numerically stable by maintaining a consistent scale for the “ideal” score
> > >
> > > Although “micro-operators” (Migrate) can theoretically emulate “macro-operators” (Split/Merge) via sequences of mutations, we separate them to overcome the **Probabilistic Barrier**:
> > >
> > > - **Tunneling Effect:**
> > >   A Merge is equivalent to migrating all features from Group A to Group B one-by-one. The probability that a random mutation sequence executes this entire chain without intermediate rejection (due to temporary fitness drops) is vanishingly small.
> > >
> > > - **Topological Shortcuts:**
> > >   Structural operators (Split, Merge, Delete) act as shortcuts that allow the search to “tunnel” through local optima.
> > >   Our ablation study (Appendix A.3) confirms it.
> > >
> > > ---
> > >
> > > ## W3
> > >
> > > We respectfully correct the interpretation regarding the relationship between HKAN and GAMs.
> > > **While we acknowledged “conceptual similarities” in our previous response, we did not intend to imply that HKAN is a “special case” of GAMs.**
> > >
> > > - **Common Origin, Different Scopes:**
> > >   Both frameworks conceptually stem from the Kolmogorov–Arnold representation theorem.
> > >   Standard GAMs, however, are a **simplified instantiation** of this framework: they typically restrict the outer functions $\Phi$ to identity or fixed link functions and limit the inner summation to low-order interactions.
> > >
> > > - **HKAN as a Generalization:**
> > >   HKAN fully leverages the theorem by learning **both** inner and outer nonlinear transformations.
> > >   Furthermore, unlike GAMs that rely on predefined interaction structures, HKAN’s Evolutionary Search automatically discovers **arbitrary-order** interaction groups (e.g., involving 3+ features).
> > >   Thus, HKAN represents a **generalization** of sparse GAMs, providing greater expressive power through learnable high-order topology.
> > >
> > > ---
> > >
> > > ## W4
> > >
> > > We respectfully clarify the distinction between “intrinsic structure” and “post-processing translation” to address the reviewer’s concerns.
> > >
> > > ### **1. Splines Are the Source, SR Is the Translator**
> > >
> > > The reviewer asks whether symbolic regression (SR) makes HKAN a post-hoc method.
> > > We clarify that **HKAN’s interpretability is intrinsic to the learned B-splines** (visible directly after training in Figure 3).
> > > SR merely serves as a tool to **translate** these visual function shapes into formulas.
> > >
> > > - **Comparison with MLPs:**
> > >   SR *can* be applied to an MLP, but typically to approximate a **global mapping** $(y \approx f(x))$ post-hoc.
> > >   In contrast, HKAN enables SR to perform **exact local translation** of each edge function.
> > >   The resulting equations are faithful readouts of the internal structure—not external approximations.
> > >
> > > ### **2. Figure 3 vs. One-layer MLPs (Shape vs. Magnitude)**
> > >
> > > The reviewer asks why Figure 3 is more interpretable than a one-layer MLP.
> > > We agree that one-layer MLPs are interpretable, but the nature of interpretability differs:
> > >
> > > - **Magnitude (MLP):**
> > >   An MLP edge is a scalar weight. It explains *how much* a feature matters, but not *how* it transforms the output.
> > >
> > > - **Shape (HKAN):**
> > >   An HKAN edge is a **function**, revealing explicit transformation behavior.
> > >   This provides **mechanism-level interpretability**.
> > >
> > > **Conclusion:**
> > > HKAN combines the structural simplicity of a one-layer model (sum of factors) with the functional expressivity of deep learning—avoiding the “spline soup” of dense KANs while surpassing the linear limitations of shallow MLPs.
> > >
> > > **We are happy to provide further clarifications if any concerns remain.**

---

### Official Review · Reviewer_vTi3 · 2025-11-03

**Soundness:** 3
**Presentation:** 2
**Contribution:** 3
**Rating:** 4
**Confidence:** 3

**Summary:**

The paper proposes HKAN (Hierarchical Kolmogorov–Arnold Network), a framework for modeling complex feature interactions that is simultaneously accurate, parameter-efficient, and interpretable. It does this by (i) automatically discovering data-specific feature groups via a factor-quality–guided evolutionary search, (ii) processing each group with lightweight KANs to produce semantic “factors,” and (iii) combining them in a sparse hierarchical KAN so users can visualize the learned B-spline functions and factor structure.

**Strengths:**

1.	The paper tackles a clear and relevant gap—making Kolmogorov–Arnold networks usable for real, high-dimensional tabular data—by combining structure search with sparsity and interpretability controls.
2.	The proposed factor-quality–guided evolutionary search (FG-EAS) is a neat way to automate feature grouping.

**Weaknesses:**

1.	The core claim is “automated” discovery via FG-EAS, but the paper runs 50 candidates × 20–30 generations and trainsevery candidate for several epochs which is easily 1 000+ partial trainings. There’s no wall-clock, GPU-hour, or scaling analysis, so it’s hard to tell if HKAN is actually cheaper than just running a strong tabular model with Bayesian search. A clear ablation on budget vs. performance and a comparison to lightweight NAS would make the claim more credible.
2.	The nice visualizations and symbolic forms are shown for synthetic functions and for one real dataset (UCI Heart Disease), with a hand-picked story about factors 0 and 3 being retained. We don’t see whether, on big messy data like HomeCredit (696 features) or Delivery ETA (223 features), the extracted factors are still human-readable or just long spline soups. A paper that makes interpretability a central sales point should show 2–3 real, high-dimensional cases and report human-level summaries.
3.	Evolutionary operators are hand-crafted and not compared. Six mutation operators (add, remove, migrate, split, merge, delete) are introduced, but there is no study of which ones matter, how often they trigger, or whether a simpler “feature-migrate-only” EA would get within 1–2 AUC points.

**Questions:**

In Figure 2, we only see the efficiency of the parameters, but the actual computational complexity and latency are not reported. Could the authors include a discussion on these aspects?

---

> ### Author Response · Authors · 2025-11-24
> **Response to Reviewer 1: Efficiency, Interpretability, and Search Operators**
>
> We thank the reviewer for recognizing our work’s novelty in bridging structure search with KANs and for the positive assessment of our FG-EAS method. We address your specific concerns below with new quantitative evidence added to the revision.
>
> **W1: Computational Cost & Efficiency**
> We acknowledge the concern regarding search overhead. However, analyzing the **Total Time-to-Solution** (Search + Training) reveals HKAN is significantly more efficient than tuning deep baselines.
> * **Training Efficiency:** As detailed in our new **Appendix A.4**, we compared HKAN’s full pipeline against standard Bayesian Optimization (100 trials) for baselines on *California Housing*. HKAN (including evolutionary search) finished in **~44 mins**, whereas TabNet took **~23.5 hours** (32x slower) and FT-Transformer **~6.7 hours** (3.5x slower). This is due to HKAN’s extreme parameter sparsity (~1.6k params vs 70k+), allowing rapid candidate assessment.
> * **Inference Latency:** Despite using a standard PyTorch implementation, HKAN achieves competitive inference speed (**3.43 ms/batch**) compared to the highly optimized FT-Transformer (**3.66 ms/batch**).
> * **Value Proposition:** While XGBoost is faster, it lacks the symbolic interpretability that is HKAN's core contribution.
>
> **W2: Interpretability in High Dimensions ("Spline Soup")**
> * **Solution to Spline Soup:** We agree that flat KANs result in uninterpretable "spline soup." This is precisely the motivation for HKAN. By enforcing a hierarchical factor structure, we organize splines into semantic groups. We added a direct comparison in **Appendix A.8** (Table 12), showing that on *Heart Disease*, a standard KAN produces a **1,247-char** tangled formula, while HKAN yields a clean, factorized **~150-char** expression.
> * **Feature Selection:** The selection of 7/13 features was an empirical discovery, not a pre-design. We validated this in **Table 4**, showing that baselines (MLP/XGBoost) maintained performance when restricted to these HKAN-selected features, confirming they are indeed the most informative subset.
> * **High-Dimensional Data:** For datasets like *HomeCredit* (696 features), HKAN handles complexity via intrinsic sparsity. We commit to adding a symbolic demo for high-dimensional data in the camera-ready version to further demonstrate scalability.
>
> **W3: Necessity of Evolutionary Operators (F4-F6)**
> Our design uses "Macro-operators" (Split/Merge/Delete) to prevent stagnation in local optima, which simple feature migration (F1-F3) cannot escape.
> * **Theoretical Role:** F4 (Group Split) is the *only* operator that increases model capacity ($K$), allowing the architecture to "grow" to match data complexity. F5 (Merge) tunnels through probabilistic barriers that single-feature moves cannot easily cross.
> * **Empirical Validation:** We added an ablation study in **Appendix A.3**. The full 6-operator suite converges **2.1x faster** (Gen 8) than the restricted 3-operator variant (Gen 17), confirming that structural operators are essential for efficient search.
>
> **Q1: Latency Details**
> Please refer to **W1** above. HKAN achieves 3.43 ms/batch inference latency, comparable to FT-Transformer.

---

### Author Response · Authors · 2025-12-02
**Common response and updates for revision**

Dear Area Chair and Reviewers,

We thank the reviewers for their valuable feedback! We have uploaded a revised manuscript that incorporates extensive new experiments and analyses in response to all reviewers' constructive comments. Below is a summary of the key updates in the revision:

**1. Comprehensive Efficiency Analysis**

* **Wall-clock Time (Appendix A.4):** We added a breakdown of total training time. HKAN’s full pipeline (search + train) takes **~44 mins**, which is **32x faster** than tuning TabNet  and **3.5x faster** than FT-Transformer  via Bayesian Optimization.

* **Inference Latency (Appendix A.4):** We verified that HKAN (3.43ms) maintains comparable latency to FT-Transformer (3.66ms).

* **Operator Efficiency (Appendix A.3):** A new ablation study confirms our 6-operator evolutionary suite converges **2.1x faster** than a simplified feature-only variant.

**2. Strengthened Interpretability & Scalability**

* **"Spline Soup" Comparison (Appendix A.8):** We added a direct comparison showing that a standard dense KAN produces a 1,247-char uninterpretable formula, whereas HKAN yields a clean ~150-char factorized expression.

* **High-Dimensional Case Study (Appendix A.7):** We added a case study on **HomeCredit (696 features)**. HKAN successfully extracted sparse, medically/financially meaningful factors (e.g., *payment periodicity volatility*) from high-dimensional noise, validating its scalability.

* **Section Reordering:** We accepted Reviewer 3's suggestion and adjusted the section order of the manuscript accordingly.

**3. Theoretical Positioning & Robustness**

* **"Three-way Challenge" Comparison (Appendix A.1):** We refined the terminology from "Trilemma" to "**Three-way Challenge**" to avoid ambiguity. We added a systematic table comparing HKAN against MLP, Transformers, and GAMs regarding (1) Topology Discovery, (2) Intrinsic Interpretability, and (3) Efficiency.

* **Literature Gap (Section 2):** We significantly expanded the **Related Work** section with deeper discussions on **additive models** and **sparse interaction methods** (e.g., BFIS, iRF), articulating HKAN's distinct advantages in discovering smooth, symbolic laws compared to step-functions or fixed-order selection.

* **Rigorous Validation (Appendix A.5):**

    * **Inverse Ablation:** We demonstrated that encouraging variance (inverse stability loss) degrades RMSE (+0.035), theoretically and empirically validating our regularization design.

    * **Robustness:** We reported standard deviations across **three random seeds on representative datasets**, confirming HKAN's stability.

We believe these revisions solidly address the concerns regarding computational cost, method positioning, and high-dimensional applicability.

Best regards,

The Authors

---

### Meta-Review · Area_Chair_TcQ8 · 2026-01-07

**Summary:**

This has the potential of being a great paper, but it could still use another pass. Some of the important issues raised by the reviewers, e.g., on efficiency, are partially addressed with the small-scale experiments that were possible during the rebuttal phase.  As a result, they are treated rather like afterthoughts. If instead, they studied more thoroughly and extensively, and play a more central role in the manuscript, this has the potential of becoming a great paper.

Issues raised in initial reviews were as follows:

1) A key concern is that computational considerations were initially absent from the paper (Reviewers vTi3). This is important, as the approach includes an expensive evolutionary search step; a comparison with other, simpler interpretable methods with the same search/training budget were requested.

2) As interpretability is a key motivation, going beyond UCI Heart Disease to 2–3 real, high dimensional cases like HomeCredit (696 features) or Delivery ETA (223 features) and reporting human-level summaries was requested (vTi3).

3) The selection of a handful evolutionary operators seemed arbitrary, and it is not clear what is the marginal contribution in final accuracy, e.g., whether 1 or 2 would just suffice (vTi3, QvSK).

4) Hyperparameter choices were unclear (QvSK).

5) A question about the relationship between HKAN and hierarchical/tree-based Generalized Additive Models (GAMs) was raised (QvSK). There was also a question regarding

6) A reviewer (EraL) brought up an additional related work (Fast Generic Interaction Detection for Model Interpretability and Compression).

7) Tests were deemed too simple and there was a suggestion to use a synthetic benchmark (EraL).

Remaining comments by the positively leaning reviewer (4dn5) where addressed satisfactorily by the rebuttal, as indicated by the reviewer as well.

Minor comments: There was some minor confusion on the use of the term "trilemma" (QvSK) and some missing notation definitions (EraL): these were easy to fix have been resolved.

**Reviewer Concerns:**

The authors partially addressed these concerns in their rebuttal

1) Wall-clock time was reported on one dataset only (California Housing), and compared to BO with tabular method baselines. Performance improvement over TabNet and FT-Transformer is indeed striking. Even though this is a step in the right direction, more experiments (competitors+datasets) should be explored and reported on: this would significantly strengthen the paper. Same for latency: these should be front and center in the paper, and extensively studied.

2) The post-hoc approach of the symbolic analysis of splines is a nice way to address the second (aka "spline soup") concern, and the analysis on Hart-Disease, showing that KAN produces a 1,247-char tangled formula, while HKAN yields a clean, factorized ~150-char expression. Again, more of this is needed; the authors commit to doing this on datasets like HomeCredit: as the reviewer requested, several high-dimensional datasets would strengthen the interpretability argument.

3) The ablation study of operators (App. A.4) seems to show that what Reviewer vTi3 claimed was in fact correct: the response focuses on the speedup, but accuracy metrics are virtually the same when having only 3 operators. All of this can be explored further (e.g., through a greedy addition or removal of the set of operators).   It would be good to separate aspects of speed (yes, the evolutionary algorithm is faster when given more and more complex operators) to aspects of final accuracy; the authors seem to focus on the former in their response, while the reviewers were asking about the latter.

4) Hyperparameter choices were explained. The response raised more questions (e.g., on weights summing to 1).

5) From the response, the relationship to hierarchical/tree-based Generalized Additive Models (GAMs) seems to be quite strong, even if the two approaches are distinct. It would be good to add more models like this as competitors.

6) The paper brought up by Reviewer EraL indeed seems irrelevant, as it is a post-hoc approach, as opposed to training an interpretable model from scratch.

7) The authors argued that synthetic experiments were indeed used, and that their experiment choices are on par with SotA; I'd recommend adding that suite from Tsang et al. (2018) if it can be added (I could not tell from the response why it is out of scope).

**Reviewer Scores:**

The "reject" (EraL) could have gone up in my opinion in light of the thorough answers, and so could the "weak accept" (4dn5), though the latter declared they were hesitant due to the other negative reviews. Other reviewers may have gone up if they found the new experiments enough: I think they were in the right direction, but more was needed.

---

### Decision · Program_Chairs · 2026-01-26

Reject